# Attractiveness of female sexual signaling predicts differences in female grouping patterns between bonobos and chimpanzees

Martin Surbeck [1,2,9✉], Cédric Girard-Buttoz [2,3,4,9✉], Liran Samuni[1,3], Christophe Boesch [2], Barbara Fruth [5,6,7], Catherine Crockford[2,3,4], Roman M. Wittig [2,3,8] & Gottfried Hohmann[2,7]

Here we show that sexual signaling affects patterns of female spatial association differently in chimpanzees and bonobos, indicating its relevance in shaping the respective social systems. Generally, spatial association between females often mirrors patterns and strength of social relationships and cooperation within groups. While testing for proposed differences in female-female associations underlying female coalition formation in the species of the genus *Pan*, we find only limited evidence for a higher female-female gregariousness in bonobos. While bonobo females exhibited a slightly higher average number of females in their parties, there is neither a species difference in the time females spent alone, nor in the number of female party members in the absence of sexually attractive females. We find that the more frequent presence of maximally tumescent females in bonobos is associated with a significantly stronger increase in the number of female party members, independent of variation in a behavioural proxy for food abundance. This indicates the need to look beyond ecology when explaining species differences in female sociality as it refutes the idea that the higher gregariousness among bonobo females is driven by ecological factors alone and highlights that the temporal distribution of female sexual receptivity is an important factor to consider when studying mammalian sociality.

[1] Harvard University, Department of Human Evolutionary Biology, Cambridge, MA, USA. [2] Max Planck Institute for Evolutionary Anthropology, Leipzig, Germany. [3] Taï Chimpanzee Project, Centre Suisse de Recherches Scientifiques, Abidjan, Ivory Coast. [4] Institut des Sciences Cognitives, CNRS, Lyon, France. [5] Faculty of Science, School of Biological and Environmental Sciences, Liverpool John Moores University, Liverpool, UK. [6] Centre for Research and Conservation, Royal Zoological Society of Antwerp, Antwerp, Belgium. [7] Max Planck Institute of Animal Behavior, Konstanz, Germany. [8] German Centre for Integrative Biodiversity Research (iDiv) Halle – Jena – Leipzig, Leipzig, Germany. [9] These authors contributed equally: Martin Surbeck and Cédric Girard-Buttoz. ✉email: msurbeck@fas.harvard.edu; cedric.buttoz@gmail.com

Group-living entails costs and benefits to each individual. Costs of sociality include increased feeding and mating competition and the risk of disease and parasite transmission[1–3]. Yet animals can also derive benefits from living in a group in the form of lower predation risk, more intense territory and resource defense against other groups as well as facilitated access to mating partners[2,4,5]. Within a group, the spatial association between individuals often mirrors the pattern and strength of social relationships[6] and strong social relationships and frequent spatial association promote within-group cooperation[7–11]. Variation in association patterns within groups has therefore been used to make inferences about several aspects of social life including reproductive strategies and potential cooperation between individuals[12,13].

While most social species live in stable cohesive social groups, some species exhibit a high degree of fission-fusion dynamics in their grouping patterns[14]. In such species, individuals often range in subgroups varying in size, composition, and duration (also called parties), allowing members of a group to adapt to environmental changes including the availability of resources while concurrently maintaining a certain degree of association[15,16]. Species with a high degree of fission-fusion dynamics offer the opportunity to assess individual partner preference through spatial associations.

Among the species with a high degree of fission-fusion dynamics are bonobos (*Pan paniscus*) and chimpanzees (*Pan troglodytes*)[17,18], which often serve as a referential model for our own evolutionary processes. They share many social and physical traits, such as multi-male multi-female communities with male philopatry and typical female dispersal, a moderate sexual dimorphism with females being smaller than males and females exhibiting sexual swellings indicative of the likelihood of ovulation[19–23]. However, bonobos and chimpanzees differ in some fundamental social aspects such as the frequency and form of female and male cooperation, intersexual dominance relationships, and the intensity of male-male competition reviewed in[24]. In bonobos, most cooperation in agonistic contexts occurs among females[25], most of whom outrank all adult males in the group and hardly receive coercive aggression by males[26]. Bonobo males cooperate mostly with females and exhibit low levels of aggression compared to chimpanzee males[27–29]. Cooperation among chimpanzee females is evident although less frequent in comparison to other sex dyads[30,31]. Chimpanzee females are all subordinate to males[32] and receive relatively high rates of coercive aggression, especially in some eastern chimpanzee populations[33]. Chimpanzee males collaborate in contexts such as territorial defense and hunting but compete aggressively within communities for rank and mating opportunities[30,34–36]. Whereas males are the ones most systematically engaging in these large-scale collaborative activities, in some communities, and especially in the Western chimpanzees, females often join such activities such as border patrols and aggressive encounters with other groups[36,37].

A general framework explaining the underlying conditions resulting in these species differences is derived from classic theories on primate socioecology[38–42]. They propose that ecological factors such as reduced seasonality[43], increased food abundance, and/or a higher abundance of terrestrial herbaceous vegetation in the diet of bonobos in comparison to chimpanzees[44] lower the degree of female-female feeding competition, allowing bonobo females to be more social and cooperative than chimpanzee females[45].

One hypothesized process driving the variation in sociality in chimpanzees and bonobos that may result in species difference in female associations, is directly related to this framework and emphasizes only aspects of the ecology. According to this first hypothesized process, a more even distribution of food resources, higher availability of food while traveling, and/or an increased food abundance decreases the cost of association for bonobo females as compared to chimpanzee females. This would result in a generally higher female-female gregariousness in bonobos by the virtue of ecology alone (assuming comparable benefits of sociality) allowing for the establishment of stronger bonding and cooperation among females. We refer to this process as the "ecology process". While previous studies have established the importance of ecological factors in driving the grouping patterns of both bonobos and chimpanzees e.g.,[46–50], direct comparisons of food abundance between selected bonobo and chimpanzee sites do not find consistent support for the basic premise of habitat differences e.g.,[51]. Furthermore, we do not know whether a fluctuation in food abundance affects female-female gregariousness similarly in both species.

A second hypothesized process, that may cause the variation in sociality in chimpanzees and bonobos, derives from theories on sexual signaling and the evolution of female sexual swellings in primates[52,53]. This second hypothesized process relates differences in sexual swellings cycles between the species to variation in behaviour and consequently to female-female gregariousness. The longer period of maximum tumescence and the more frequent and earlier swelling cycles during an interbirth interval in bonobos as compared to chimpanzees are thought to lead to extended time periods of females attracting other members of the group, including other females[20,21,45]. While in both species females exhibiting sexual swellings attract males and females[46,49,54,55], the underlying processes resulting in female-female attraction might differ in some aspects between the species. In chimpanzees, sexual attraction of maximally tumescent females may be tuned towards attracting males, and other females benefit from being in parties with males. The benefits to females can include increased predation protection, increased access to meat resulting from male hunting and better social information resulting from observing third-party interactions among males[46,56]. In bonobos, the signaling effect of maximally tumescent swellings could be similar effects[57], but there are likely additional benefits to females resulting directly from association with maximally tumescent females.

These additional benefits of female-female associations include indirect fitness benefits resulting from maternal support to adult sons during mate competition which would specifically lead to mothers of adult sons being attracted to maximally tumescent females[27] and the increased opportunities for female-female socio-sexual behaviour that facilitates female coalitions formation hypothesized to be relevant to control male aggression[25,54,58]. These benefits are specific to the nature of the social relationships observed in bonobos (impactful mother-son bonds and female-female cooperation facilitated by sociosexual behaviour), and likely result in self-reinforcing processes driving potentially higher females' gregariousness in this species. Given the potentially higher benefits of females associating with maximally tumescent females in bonobos, the effect of the presence of maximally tumescent females on party size should be larger in bonobos as compared to chimpanzees. Finally, the costs of associating with males might be lower in bonobos due to higher female dominance ranks and the reduction of male aggression against females, potentially because bonobo sexual swellings also attract females, providing opportunities for female-female alliances[26]. Again, these benefits are closely linked to the nature of the social relationships in bonobos and not necessarily to a generally higher female-female gregariousness. We refer to this process of female attraction to maximally tumescent females as the "sexual signaling attraction process", even though differences in the ecology likely affect sexual signaling by females in both *Pan*

species[45,52]. Yet the immediate effect of sexual signaling in itself in both species can be established when controlling for variation in ecological factors such as food abundance.

Understanding how these two hypothesized processes account for variation in female sociality within and between species potentially allows us to identify proximate processes driving the observed species differences in female, and potentially, male behaviour, and to make inferences about potential mechanisms resulting in the divergent evolution of bonobos and chimpanzees.

Though general differences in female gregariousness of the two species are widely accepted[59], this is mostly based on comparisons of published datasets collected under different protocols[49,60]. These comparisons show that, compared to chimpanzees, bonobo female associations are larger and include a larger proportion of females from the community[60], and that mixed-sex associations are more frequent[29,49]. However, there appears to be a large variation in female gregariousness across chimpanzee populations, with Taï Western chimpanzees appearing more gregarious than some chimpanzee populations, in particular Eastern chimpanzees[50,55,61]. Nevertheless, the chimpanzees at Taï exhibit the typical chimpanzee patterns of behaviour, including preferred association with same-sex individuals in parties[29], male dominance over females, and a frequent occurrence of male aggression against females[32].

Generally, standardized assessments of party sizes across populations and species are essential to comparatively test hypotheses explaining variation in grouping patterns. Studies often vary in their focus (e.g. focus on both sexes or one sex only), in the spatial and temporal definitions of parties e.g.,[62,63] and in the methods of data collection (focal individual vs focal party[64]), all of which likely bias the outcomes. For example, the focal party follows, as compared to focal individual follows, likely underestimate the times spent alone by individuals. Furthermore, female gregariousness is sometimes inferred from studies that focus on males[65] or group size estimates based on nest counts[51] which is unlikely to lead to reliable estimation of this parameter. Different studies also differ in the way they account for party membership ranging from a continuous way of keeping track of all individuals in sight at a given time[56] to a cumulative way to sum the presence of individuals seen over a defined time span e.g.,[66], the latter potentially resulting in higher numbers (discussed in[67]). The choice of methods is often influenced by the visibility of the environment with environments with poorer visibility using the cumulative way. Furthermore, observers often follow focal parties, which likely result in observations biased towards larger, noisier, more stationary associations of individuals and a decreased probability to detect solitary individuals, which could be particularly prominent in the less conspicuous bonobos[60]. Finally, the species comparisons published to date do not incorporate differences in female sexual signaling, which limits assessments of the mechanisms underlying female and intersexual grouping dynamics.

The lack of comparable data on gregariousness between bonobos and chimpanzees hinders tests of the predictions of potential evolutionary scenarios resulting in the differentiation of behaviour in these two species. In this study, we therefore applied an identical methodology of data collection to quantify the drivers of female-female gregariousness in two wild populations, bonobos at LuiKotale in the Democratic Republic of Congo and chimpanzees in the Taï forest in Ivory Coast. Specifically, we first tested for the described species differences in female-female gregariousness between three similar-sized groups, the Bompusa community in LuiKotale and the East and South communities in Taï. Secondly, we tested how variation in female sexual signaling and a behavioural proxy for food abundance, the percentage of time feeding, explained the observed within and between species variation in female-female associations. This proxy for food

abundance is based on a range of studies showing that feeding time decreases when food abundance or quality increases in several primate species, including Taï chimpanzees[68–73]. Furthermore, using unpublished data, we can show the same pattern in wild bonobos (see Supplementary Information).

If generally higher gregariousness in bonobos than in chimpanzees promotes the differences in social behaviour, including patterns of female cooperation, aggression, and dominance between the sexes (as hypothesized by the ecology process), we would expect (1) a larger number of females in parties of focal females (Prediction P1) and (2) less time spent alone by females in LuiKotale bonobos compared to Taï chimpanzees (Prediction P2).

To specifically test whether the more favorable ecological conditions in bonobos are the main drivers of the hypothesized species difference in female-female gregariousness between chimpanzees and bonobos (ecology process), we first focused on situations in which attraction towards sexually signaling females could be ruled out. We predicted that bonobo females have generally a higher number of female party members (Prediction P3a) and spend less time alone (Prediction P3b) than chimpanzee females in the absence of maximally tumescent females (MTFs). Second, we focused on a situation in which male-female (i.e. mate) attraction could be ruled out, by using all female parties only, thus excluding mate attraction. We predicted that bonobo females have a higher tendency to associate with other females (i.e. spend less time alone) in the absence of males as compared to chimpanzee females (Prediction P4). Finally, we also hypothesized that if the ecology process applies, the hypothesized effect of the presence of MTFs on female-female gregariousness (see below) is mostly driven by ecology so that favorable ecological conditions trigger sexual signaling and the effect of sexual signaling is simply a by-product of the recent ecology. Here we specifically predicted that, after controlling for temporal monthly fluctuations in food availability, we would not find the effect of the presence of MTFs on gregariousness (Prediction P5).

Alternatively, if ecological conditions are generally not more favorable in bonobo habitat or if the outlined ecology process is not the main proximate driver of female-female gregariousness in *Pan*, species difference in gregariousness could be mostly driven by the extended duration and the higher attractiveness of sexual swelling in bonobo females as compared to chimpanzees (sexual signaling attraction process). If, as predicted by this hypothesis, MTFs attract more females to join their party and/or MTFs actively seek to range in larger parties, we predict a positive effect of the presence of maximally tumescent females on female-female gregariousness in both species even after controlling for the effects of fluctuation in food availability (Prediction P6). We predict, as previously reported[21], a higher number and a more frequent presence of maximally tumescent females in bonobos (Prediction P7) leading to an overall higher number of females in female-female parties in bonobos. This effect would be reinforced if the effect of the presence of MTFs on female-female gregariousness is stronger in bonobos as compared to chimpanzees (Prediction P8). Finally, if the sexual signaling attraction process is the main proximate driver of female-female gregariousness we also expect no species differences in measurements of female gregariousness in the absence of maximally tumescent females between chimpanzees and bonobos (Prediction P9).

Unlike previous comparative studies, we only found limited evidence for generally higher gregariousness of bonobo females as compared to chimpanzee females when comparing association patterns between two chimpanzee and one bonobo communities with similar numbers of females. While bonobo females had on average a slightly but significantly higher number of females in their parties as compared to their chimpanzee counterparts, this

**Table 1 Overview over structure and results of the models analyzing differences in female-female gregariousness between one bonobo (Bompusa) and two chimpanzee communities (Taï East and Taï South).**

| MODEL | Model 1: Average female party size $N$ half-days = 409, $N$ females = 34 | | | | | |
|---|---|---|---|---|---|---|
| Response | Average number of females in party over the course of a half-day focal follow | | | | | |
| Full-null model | LRT, df = 2, $\chi^2$ = 6.15, P = 0.046 | | | | | |
| | Est. | SE | $CI_{low}$ | $CI_{high}$ | $\chi^2$ | P |
| Intercept | 1.52 | 0.07 | | | | |
| Community (Taï_East) | −0.24 | 0.10 | −0.44 | −0.06 | 6.15 | 0.046 |
| Community (Taï_South) | −0.14 | 0.09 | −0.30 | 0.01 | | |
| Time of the day (AM) | −0.03 | 0.06 | −0.15 | 0.08 | 15.47 | <0.001 |
| Time of the day (AM + PM) | −0.40 | 0.11 | −0.63 | −0.20 | | |
| Random factors | Focal identity | | | | | |

difference disappeared when we compared female-female gregariousness between the two species during times when no sexually attractive (maximally tumescent) females were present. Furthermore, the percentage of time spent alone, and the time spent in female-only parties were comparable between the two species. Our findings do not support the idea of an "ecology process" underlying female-female grouping patterns, which assumes that bonobo females are generally more capable to aggregate with each other in large numbers than chimpanzee females by the virtue of their ecology alone. As expected from the described difference in sexual signaling between the species, we found that maximally tumescent females were more frequently present in the parties in bonobos than in chimpanzees. Furthermore, when controlling for food availability, we found that in bonobos the presence of maximally tumescent females is associated with a stronger increase in female-female gregariousness than in chimpanzees. This is in line with predictions of the female sexual signaling process which attributes differences in female associations directly to factors related to the sexual signaling. Given the findings of very small numerical differences in overall female-female gregariousness, it seems unlikely that current selection favoring a generally higher female affinity in bonobo parties is ultimately driving the proposed species differences in female cooperation that have been attributed to female dominance and reduction of male coercive aggression. The significant species differences in female-female gregariousness match differences in sexual signaling and in the female affinity to parties with potentially fertile females. Therefore, our results support the idea that changes in female signaling are a proximate driver of species differences in social structures with behavioural implications for relationships between and within each sex.

## Results
We conducted a total of 409 half-day focal follows on adult females ($N_{bonobo}$ = 175, $N_{Taï\ East}$ = 110, $N_{Taï\ South}$ = 124; Supplementary Table 1) with an average duration of 4.9 h of observation per follow ($N_{bonobo}$ = 4.5 h, $N_{Taï\ East}$ = 5.2 h, $N_{Taï\ South}$ = 5.2 h; Supplementary Table 1).

**Female party size and time spent alone.** The full-model investigating differences in the number of female party members (Model 1) differed from the null model (Table 1; LRT, df = 2, $\chi^2$ = 6.15, P = 0.046), indicating that community had an influence on the number of female party members. While there was only a trend for statistical differences between bonobo females and chimpanzee females from Taï South (4.5 vs. 3.7, p = 0.097,

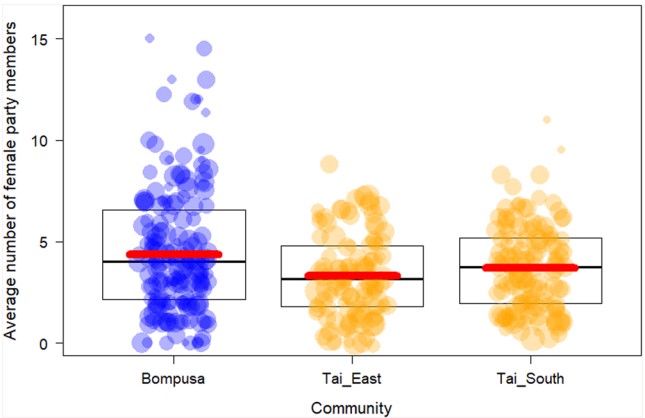

**Fig. 1 Differences between the Bompusa bonobo community and the two Taï chimpanzee communities in the total number of adult females in the party of female focal individuals.** The Bompusa bonobo community is depicted in blue and the two Taï chimpanzee communities are depicted in orange. Each dot represents a half-day focal follow and the size of the dots represents the number of data points (i.e. number of 30 min parties) for a given value. Boxes indicate the medians and the 25 and 75% quartiles. Bold red lines represent the model line controlling for time of the day and multiple sampling of the same individuals (Model 1).

**Table 2 Comparison of female association patterns one bonobo (Bompusa) and two chimpanzee communities (Taï East and Taï South).**

| Community | Bompusa | Taï East | Taï South |
|---|---|---|---|
| Time alone (% of obs. Time, averaged over females) | 6.21 | 4.08 | 7.18 |
| Time in same-sex parties (% of obs. time) | 15.93 | 20.46 | 34.21 |
| Number of females in party when MTF absent | 3.21 | 2.99 | 3.55 |

MTF refers to "Maximally tumescent females".

Fig. 1), female bonobos had more females in their party than chimpanzee females in Taï East (4.5 vs. 3.3, p = 0.01, Fig. 1). Visual inspection of individual means indicates that bonobo females with an adult son in the community had the largest average female party sizes (Supplementary Fig. 2).

The full-model investigating differences in the amount of time spent alone by females (Model 2a) was not significantly different from the null model (Table 2; LRT, df = 2, $\chi^2$ = 1.19, P = 0.55), indicating that community had no significant influence on the time spent alone (Supplementary Fig. 2A). Visual inspection of individual means indicates that females with an adult son in the community spent the least time alone in both species, which was potentially more pronounced in bonobos (Supplementary Fig. 2).

Bonobo and chimpanzee females also differed neither in the percentage of time spent alone during times when they were not associated with males (Model 2a; LRT, df = 2, $\chi^2$ = 1.19, P = 0.55) nor in the time they spent in all female parties (Model 2b, Table 2; LRT, df = 2, $\chi^2$ = 1.02, P = 0.599, Supplementary Fig. 2C). Visual inspection of the average time spent in all female parties, shows that among bonobo females, mothers with adult sons seemed to spend the least time in such associations which was not the case in Taï South chimpanzees (this could not be assessed for Taï East since none of the females in this community had adult sons present, Supplementary Fig. 2C).

**Table 3 Overview over structure and results of the models analyzing differences in female association patterns between one bonobo (Bompusa) and two chimpanzee communities (Taï East and Taï South) in relation to the presence of maximally tumescent females (MTF) while controlling for monthly percentage time feeding (proxy for fluctuation in food availability) and with bonobos and no MTF present as the reference.**

| MODEL | Model 3a: Variation in female party sizes in relation to maximally tumescent females (categorical variable: 0, less and equal 1, more than 1) $N$ half-days = 409, $N$ females = 34 | | | | | | Model 3b: Variation in female party sizes in relation to maximally tumescent females (percentage time with maximally tumescent females) $N$ half-days = 409, $N$ females = 34 | | | | | |
|---|---|---|---|---|---|---|---|---|---|---|---|---|
| Response | Average number of females in party over the course of a half-day focal follow | | | | | | Average number of females in party over the course of a half-day focal follow | | | | | |
| Full-null model | LRT, df=10, $\chi^2$ = 70.03, $P < 0.001$ | | | | | | LRT, df=7, $\chi^2$ = 56.83, $P < 0.001$ | | | | | |
| | Est. | SE | CI$_{low}$ | CI$_{high}$ | $\chi^2$ | $P$ | Est. | SE | CI$_{low}$ | CI$_{high}$ | $\chi^2$ | $P$ |
| Intercept | 1.07 | 0.08 | | | | | 1.34 | 0.07 | | | | |
| Community (Taï East) | 0.04 | 0.11 | −0.19 | 0.24 | | | −0.17 | 0.10 | −0.37 | 0.03 | | |
| Community (Taï South) | 0.20 | 0.10 | 0.00 | 0.38 | | | −0.01 | 0.09 | −0.18 | 0.16 | | |
| MTF Cat 1 | 0.35 | 0.11 | 0.13 | 0.56 | | | | | | | | |
| MTF Cat 2 | 0.95 | 0.08 | 0.78 | 1.13 | | | | | | | | |
| Percentage time with MTF (incl focal) | | | | | | | 0.34 | 0.04 | 0.26 | 0.38 | | |
| Monthly percentage time feeding | 0.12 | 0.04 | 0.03 | 0.12 | | | 0.03 | 0.04 | −0.04 | 0.11 | | |
| Dominance rank | 0.04 | 0.04: | −0.02 | 0.11 | 1.37 | 0.29 | 0.04 | 0.04 | −0.03 | 0.11 | 0.96 | 0.327 |
| Community (Taï East): MTF Cat 1 | −0.17 | 0.18 | −0.53 | 0.15 | 20.61 | <0.001 | | | | | | |
| Community (Taï South): MTF Cat 1 | −0.27 | 0.16 | −0.59 | 0.03 | | | | | | | | |
| Community (Taï East): MTF Cat 2 | −0.76 | 0.16 | −1.11 | −0.45 | | | | | | | | |
| Community (Taï South): MTF Cat 2 | −0.61 | 0.24 | −1.15 | −0.17 | | | | | | | | |
| Community (Taï East): Percentage time with MTF (incl focal) | | | | | | | −0.23 | 0.06 | −0.35 | −0.12 | 17.25 | <0.001 |
| Community (Taï South): Percentage time with MTF (incl focal) | | | | | | | −0.25 | 0.07 | −0.39 | −0.13 | | |
| Community (Taï East): Monthly percentage time feeding | −0.34 | 0.07 | −0.47 | −0.21 | 25.59 | <0.001 | −0.26 | 0.06 | −0.38 | −0.15 | 17.10 | <0.001 |
| Community (Taï South): Monthly percentage time feeding | −0.27 | 0.08 | −0.42 | −0.12 | | | −0.19 | 0.08 | −0.33 | −0.05 | | |
| Time of the day (AM) | 0.01 | 0.05 | −0.09 | 0.12 | 11.72 | 0.003 | −0.003 | 0.06 | −0.10 | 0.1 | 11.57 | 0.003 |
| Time of the day (AM + PM) | −0.30 | 0.10 | −0.51 | −0.11 | | | −0.31 | 0.10 | −0.52 | −0.13 | | |
| Random factors | Focal identity | | | | | | Focal identity | | | | | |

**Female party size in relation to sexual swellings and fluctuations in food abundance.** The full-models investigating community differences in the number of female party members in relation to the presence of maximally tumescent females (MTFs) while controlling for fluctuations in food availability differed significantly from the null-models both when integrating MTF categories as a factor (Model 3a) and the percentage of time spent in the presence of at least one MTF (percentage MTF presence, Model 3b) (Table 2; LRT $_{MTF\ categories}$, df = 10, $\chi^2$ = 70.32, $P < 0.001$; LRT $_{percentage\ MTF\ presence}$, df = 7, $\chi^2$ = 56.83, $P < 0.001$). Furthermore, the interaction between the categorical variable for the presence of MTF and community in Model 3a and the interactions between the percentage of time spent with at least one MTF in the party and community in Model 3b were both significant (categorical variable Model 3a: df = 4, $\chi^2$ = 20.61, $P < 0.001$; percentage time: df = 2, $\chi^2$ = 17.24, $P < 0.001$, Table 3) indicating that the presence of maximally tumescent females had a different influence on female grouping patterns in the three communities.

To test prediction P5, we quantitatively compared the presence of MTF in our three study communities. Overall, MTFs were present on more days in bonobos as compared to chimpanzees (38% of observation days versus 34% $_{Taï\ East}$ and 23% $_{Taï\ South}$) and the average number of MTFs in parties were about two to five times higher (0.90 vs. 0.42 $_{Taï\ East}$/0.17 $_{Taï\ South}$). When the focal female was maximally tumescent, we observed the average number of female party members per half-day changing across days in bonobos from 4.13–6.54 ($N_{MTF\ focal}$ = 28), in chimpanzees from Taï South from 3.73–3.51 ($N_{MTF\ focal\ follows}$ = 4) and in chimpanzees from Taï East from 3.31–3.57 ($N_{MTF\ focal\ follows}$ = 13).

The results from the model incorporating the presence of maximally tumescent females in the party as a categorical variable (0, less or equal to 1 excluding 0, more than 1) show that there were no species differences in the number of female party members when there were no maximally tumescent females present (Prediction P7) but a stronger increase in the number of party females in bonobos in the presence of maximally tumescent females (Prediction P6, Fig. 2). The largest species difference occurred when there were on average more than 1 maximally tumescent female present with bonobos having larger numbers of party females. In the chimpanzee community of Taï East there seemed almost no effect of the presence of maximally tumescent females when controlling with a proxy for food availability and a weak effect in the chimpanzee community of Taï South (Fig. 2).

The results from the model incorporating the presence of maximally tumescent females as the percentage of time with at least one maximally tumescent female in the party (including the focal) revealed the same patterns of species differences in the influence of maximally tumescent females (Table 3, Fig. 3). In bonobos, an increase in the percentage of time with maximally tumescent females in the party strongly increased the number of females in the party (Fig. 3). The effect was over three times larger in bonobos than in Taï East chimpanzees and eight times larger than in Taï South chimpanzees (estimates of the effect of percentage time spent with MTF for bonobos: 0.32, for Taï east 0.09 and for Taï south 0.07; Fig. 3).

Finally, we did not observe any changes in the directionality of the effect of the influence of MTF on female grouping pattern when running a model with a reduced bonobo sample size

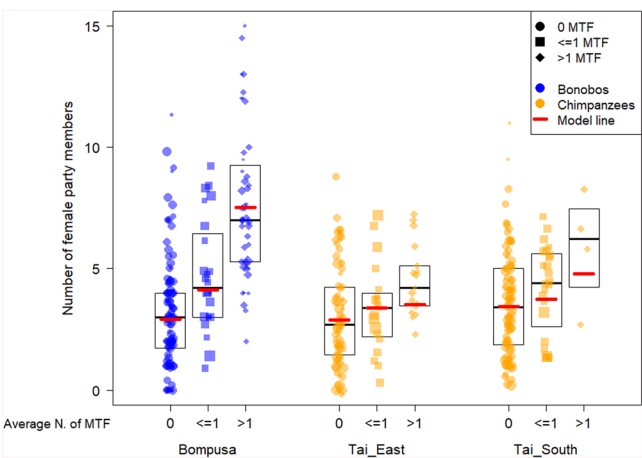

**Fig. 2 Effect of the number of maximally tumescent females (MTF) on the number of females in the party of the focal female.** The Bompusa bonobo community is depicted in blue and the two Taï chimpanzee communities are depicted in orange. 0 means no MTF, ≤1 means an average of 1 or less but not 0 MTF, and >1 means an average of more than 1 MTF were present during the focal follow [all including focal swelling]. Each dot represents a half-day focal follow and the size of the dots represents the number of data points (i.e. number of 30 min parties) for a given value. Boxes indicate the medians and the 25 and 75% quartiles. Bold red lines represent the model line controlling for food availability, dominance rank, time of the day, and multiple sampling of the same individuals (Model 3a).

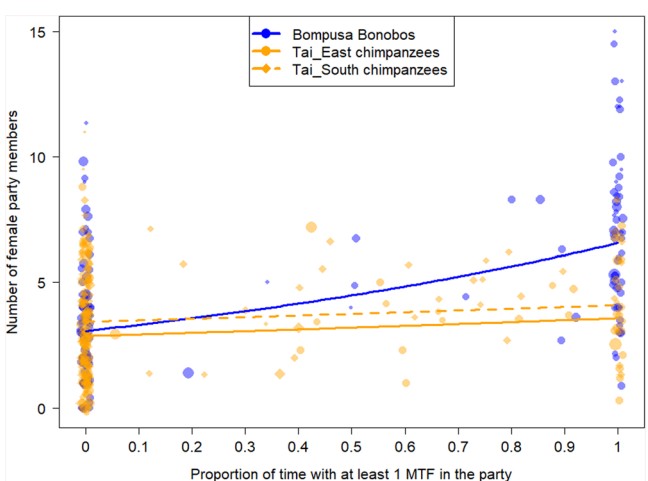

**Fig. 3 Effect of the proportion of time with at least one maximally tumescent females (MTF) during a given day on the number of females in the party of the focal female.** The Bompusa bonobo community is depicted in blue and the two Taï chimpanzee communities are depicted in orange (round dots for Taï East and diamond dots for Taï South). Each dot represents a half-day focal follow and the size of the dots represents the number of data points (i.e. number of 30 min parties) for a given value. The lines indicate the model line controlling for food availability, dominance rank, time of the day, and multiple sampling of the same individuals in solid blue, solid orange, and dashed orange for Bompusa, Taï East, and Taï South respectively (Model 3b).

including an alternative proxy for food availability (number of trees with fruits along the phenology transects for bonobos and an established food abundance index for Taï; Supplementary Table 2).

While not being an explicit part of the hypotheses testing, we inspected the influence of our proxy for food availability on female association patterns which revealed a significant species

difference (Model 3a and 3b, all $P < 0.001$, Table 3). In chimpanzees we found a negative effect, indicating that the less time they spend feeding (which occurs when there is high food availability as documented in Tai chimpanzees[72] and in wild bonobos, our study, Supplementary Fig. 1) the larger the female party size regardless of the presence of maximally tumescent females (Table 3, Fig. 3). This effect was stronger in Taï East (see Table 2, Supplementary Fig. 3). For bonobos, there was a weak positive relationship indicating a much weaker influence of food abundance on female association patterns when controlling for the presence of maximally tumescent females (Table 3, Supplementary Fig. 3). The results of a reduced dataset with phenological data revealed the same directionality of the effects (Supplementary Table 2). For a visual presentation of the changes of the number of female party members over the course of the study see Supplementary Fig. 4.

## Discussion

The discrepancy between our and earlier findings e.g.,[60] on the robustness of differences in female gregariousness between bonobos and chimpanzees can have several explanations. Firstly, it can be due to variation in female gregariousness across chimpanzee populations and the choice of our study population. The chimpanzee population of Taï is known for its high female-female gregariousness in contrast to other chimpanzee populations and consequently our data suggests more overlap in female-female gregariousness for these species than previously recognized[61,74–76]. An earlier direct species comparison of grouping patterns using the same methodology included chimpanzee populations with a lower female-female gregariousness (1.2 average female party size at Kalinzu compared to 3.4/3.7 this study[77]). The Taï chimpanzees have been traditionally attributed to the subspecies of Western chimpanzees (*Pan troglodytes verus*; but see[78]) which diverged from Central/Eastern chimpanzees around 510,000 years ago[79], and as a population might have evolved in some ways analogous to bonobos. One hypothesis for their higher gregariousness as compared to other chimpanzee populations is a higher predation pressure and in particular a high density of leopards[45,47]. Future studies directly comparing female gregariousness among chimpanzee populations with similar methodologies will inform us of how pronounced population differences are across chimpanzees. However, if we would assume that contemporary differences in female gregariousness are a crucial aspect of the divergence of bonobo and chimpanzee social behaviour, we would expect all chimpanzee populations to differ from all bonobo populations in this aspect. Comparing the average female party sizes from our study to previously published bonobo data based on party follows (which may overestimate but not underestimate average party sizes), allows us to exclude the possibility that our bonobo population has substantially smaller female party sizes (average female party members at Wamba 3.2 females[78], at Lomako 3.0[49] versus 4.5 in our study).

Secondly, methodological differences in our study compared to previous studies, namely the use of focal animal sampling instead of the party follow data have allowed us to have a better understanding of individual variation in female-female gregariousness within a population and to have a reduced observation bias towards larger parties. Observing the largest party may have overestimated female-female gregariousness in previous bonobo studies and may have led to unequal bias in bonobos as compared to chimpanzees. Furthermore, by using the same method in defining party composition in the two species, we can really compare party composition without the use of complex methods to make only aspects of grouping data comparable (e.g. social preference[29]).

Thirdly, our comparison might include data from times of special ecological conditions in the habitat of one of the species that altered female-female gregariousness as compared to the "normal" population conditions. While we do not have comparable bonobo data from earlier time periods and cannot exclude the possibility that patterns of female-female gregariousness differed from "normal" conditions, comparing the average female party sizes of chimpanzees of our study (3.4 East Group and 3.7 South Group) to earlier studies at Taï using also focal follows (3.8 South group[55]) indicates very similar results. This allows us to partially exclude the possibility, that the conditions at Taï were particularly favorable to larger parties during our study period.

Fourthly, including female sexual swellings and a proxy for food availability in the same analysis might be led to different conclusions than in previous studies. By incorporating female signaling and a proxy for food availability, we can test for the effects of female attraction towards maximally tumescent females while controlling for fluctuation in the local ecology. This allows us to assess baseline association tendencies for bonobo and chimpanzee females in the absence of sexual attraction. While our results suggest that the overall association tendencies of the females of the two species may not be fundamentally different, there are clear species differences in the dynamics of female groupings. Only bonobo female-female gregariousness seems to increase in the presence of maximally tumescent females when controlling for the effects of the ecology. While these results differ from previous chimpanzee studies reporting a stronger influence of maximally tumescent females than of ecology on party sizes[55], it must be kept in mind that these studies did not focus on female-female gregariousness but on general changes in party sizes including male party members[60].

Finally, in the absence of phenological data and data on the local abundance of bonobo-feeding trees during our study period, we controlled for fluctuations in ecology by including a behavioural proxy. We used the time spent feeding, as it has previously been shown to negatively correlate with food abundance in several great African ape populations including our chimpanzee study population[70–73]. Furthermore, we found that this relationship also applied to wild bonobo communities (Figure S1) and thus are confident that our proxy truly reflects variations in food abundance in the habitat. Given that our results remained unchanged when reducing the dataset in bonobos to months where data have been collected that allowed us to infer a general fruit availability, we are confident in our conclusion that the female sexual signaling attraction process is the most likely to explain the difference in female-female gregariousness between bonobo and chimpanzee. For future studies, the inclusion of direct measures of food availability would be preferable. Our results showing the limited influence of ecology on the bonobo association patterns seem furthermore to support previous work showing that bonobo party sizes seem rather independent of fruit availability[60]. It must be kept in mind that our proxy for ecology does not allow us to make inferences about population differences in food availability, but rather about population differences in the influence of local fluctuations in food availability on female gregariousness. To investigate population differences in the species relevant productivity of the environments other measures would have to be used.

The evolutionary origin of female-female opportunistic coalitionary support against males in bonobos is puzzling and is thought to be a strong behavioural mechanism that reinforces a social system in which females can dominate males. While chimpanzee females engage in coalitionary support for kin and bond partners[80] and in group-level cooperative acts of territorial defense[37], there is no systematic female-based cooperative system

in place that enables females to gain or maintain dominance over males[81]. One line of argument links this form of female cooperation to increased female gregariousness in bonobos. Those larger aggregations of females then supposedly allow bonobo females to foster stronger ties with each other through mechanisms such as co-feeding and food sharing, thereby promoting alliance formation which is used to control male aggression and help females to increase their dominance status[24,25,60,82]. While the causality between female gregariousness and cooperation is not firmly established, higher gregariousness of females, potentially facilitated by extended sexual signaling, might simply create an imbalance of power in favor of females. Mutual support among females has been shown to increase female dominance, for example in spotted hyenas (Crocuta crocuta)[83].

Our results seem to indicate that an increased general cohesion and attraction between females might not be the underlying species difference facilitating stronger female alliance formation in the first place, as these differences are neither very pronounced nor persisting in the absence of sexual signaling. The species differences in female gregariousness seem closely linked to species differences in patterns of female sexual signaling and the response to it by other individuals, indicating that female sexual signaling is the crucial trait setting apart the social systems of both species of the genus Pan in ways outlined below. Future studies should explore the potential pathways, by which changes in female sexual signaling is linked to the observed characteristics of bonobo female relationships in particular and of female relationships in female-dominated societies in general[83,84]. There are some observations in our results that might be helpful insights for such studies:

Firstly, our finding that bonobo but not chimpanzee females with adult sons have the highest average numbers of females in their parties (Supplementary Fig. 2A) indicates that bonobo female grouping decisions are either influenced by potential stronger gains in indirect fitness benefits through supporting their sons or reflect a stronger tendency of adult sons to associate with their mothers, which attracts other females[27]. While maternal presence during adolescence has been shown to affect male reproductive parameters in chimpanzees[85], in bonobos the presence of a mother in a party increases the likelihood of her son to mate[27] and her presence in the community increases his likelihood to reproduce more than in chimpanzees[86]. Similar effects of mother presence on the reproductive success of their sons have been described in other species including orcas[87]. Further studies should investigate how potential indirect fitness benefits of mothers influence female sociality, ranging patterns, and the integration of other females in the community, as these mothers might also be particularly attractive association partners for other females[88].

Secondly, the increase in the duration of maximal tumescence in bonobos as compared to chimpanzees may have facilitated female-female tolerance and alliance formation[58,89]. Socio-sexual behaviours between females, which are more frequent in bonobos, and mostly include maximally tumescent females[89], reduces potential anxiety in the face of competition and seem to facilitate proximity of females even during competitive situations[58]. While there is some indication that higher rates of socio-sexual behaviour occur more frequently in dyads that cooperate more[58], future studies should assess the role of female sexual signaling on the occurrence of female-female cooperation in different contexts.

While we show that species differences in sexual signaling influence female-female sociality, there also are potential implications for female-male relationships: In both, bonobos and chimpanzees, the number of males in parties increases in the presence of maximally tumescent females[46,49]. In addition, the costs of sexual attraction hypothesis relate species differences in

male behaviour towards females to differences in the duration of sexual attractiveness[20,21,45]. The potential of a male to monopolize access to a given female is lower in bonobos due to the longer period of sexual attraction, which results in a higher number of synchronously attractive females (as shown in this study) and a more confused timing of ovulation[21,90]. Extended female sexuality and decreased male monopolisation ability likely impact male competitive behaviour and the mating strategies used by primate males (e.g.[91]). Accordingly, the reduced monopolization potential of bonobo females and a smaller likelihood of a given mating to result in paternity as compared to chimpanzees, reduces the potential benefits bonobo males might gain by using aggression and increases male-female social attraction[45,92]. Similar effects have been seen in other primate species such as Muriquis (*Brachyteles arachnoides*)[93] and Assamese macaques (*Macaca assamensis*)[94]. While a decrease in the female monopolization potential also disincentivizes male investments into physical strength and consequently facilitates female dominance, further studies should invest the link between female sexuality and male strategies in bonobos.

In the light of our results, and under the assumption that current selection still maintains behavioural differences between species, it appears unlikely that generally higher gregariousness among females based solely on ecological conditions or constraints facilitates the suggested higher levels of female-female cooperation in bonobos as compared to chimpanzees. Instead, the results of our study support the view that some marked species differences in female-female gregariousness arise from variation in female sexual signaling that in itself could explain some of the observed behavioural differences. However, why bonobo females have a longer period of tumescence, why their ovulating is better concealed than in chimpanzee females and what the role is of different ecological factors in shaping these differences remains unexplained. Future research needs to focus on the advantages and selective pressures of concealed ovulation in *Pan* species and how bonobo females can afford such long tumescence periods.

Overall, differences in female sexual signaling not only affect within and between species patterns of female associations, but they also likely change male mating strategies, incentives for male aggression, and potentially increased female leverage. How the differences in association patterns, female sexuality, and female power are linked to female bonding and cooperation is not resolved. Only a few studies at this stage provide a potential link between female sexuality and female bonding and cooperation[58] or show the advantage of female bonds on cooperation[9,36,80,95,96]. However, given our results, more studies should probe new explanations for the emergence of female dominance in bonobos and female cooperation and in bonobos and chimpanzees.

## Methods
**Study communities**. We conducted the study at the field sites of LuiKotale, DRC (Fruth & Hohmann 2018) and Taï, Côte d'Ivoire (Wittig 2018), between February 2015 and December 2017 (for an exact time span of data collection periods at each site see below). We relied on female focal individual follows to reduce the bias towards larger groups stemming from party followers and be better able to quantify the durations individuals spend alone. We followed focal animals daily in two chimpanzee communities (Taï East and Taï South) and one bonobo community (Bompusa), allowing assessment of whether or not the pattern found at least in chimpanzees were population or community-specific. We included all adult parous females at the start of our study period as our focal individuals (estimated older than13 years). The East community of Taï chimpanzee consisted of 14 adult individuals (5 males, 9 females). One of the adult females could not be followed individually due to her shyness towards observers and was not included in the party composition. None of the females in this community had an adult son present. The South community of Taï chimpanzee consisted of 18–20 adult individuals (5–6 males and 13–14 females). One female and a male disappeared during the course of the study. Two females in this community had an adult son present. Finally, the Bompusa community consisted of 20–21 adult individuals (7 males and 14–15 females, one of which immigrated into the study community after the start

of the study period and was not followed as a focal individual but included in the party composition. In addition, one extra female was a temporary visitor and not included in the analysis (see Supplementary Table 1 for details on the age and observation hours of each focal female). Three females in this community had an adult son present.

**Ethical considerations**. Ethical guidelines adhere to those defined by the former Department of Primatology at the Max Planck Institute for Evolutionary Anthropology: https://www.eva.mpg.de/primat/ethical-guidelines/?Fsize=0%252527&cHash=d70e8adae0f98580648bf02d09db18d0

**Party associations and female sexual swellings**. CGB and six field assistants collected association and behavioural data from the chimpanzee communities during two field stays from Feb.-Sept. 2016 and from Feb–May 2017 and from the bonobo community during two field stays from Feb. 2015 till May 2016 and from Aug.–Dec. 2017 (with CGB dividing time between the two sites). We conducted focal follows[97] of all adult female individuals for the duration of half a day either from the time the focal left the night nest in the morning until around noon or from noon until the focal built a night nest. If the apes were followed by an observer during the entire day, the observer changed focal individuals around noon (12:00 in bonobos and 12:30 in chimpanzees due to later sunrise at Taï compared to LuiKotale). In some rare cases, the same individual was also followed the entire day (e.g. when the female was alone the entire day the observer could not switch focal at noon). While we tried to balance the number of morning and afternoon follows for each individual and to randomly choose the order in which the individuals were followed, this choice had to be adjusted to the individual presence at the sleeping site in the morning. The observation hours per individual in bonobos was 62 h on average (range 53–72.5); the observation hours per individual in chimpanzees was 54 h on average (range: 16.5–109.5, see Supplementary Table 1 and 2). It is important to note that the focal was followed the entire half-day regardless of changes in party composition and even if the focal ended up alone, to capture true variation in association patterns. Thereby we limited to the best of our possibility the bias towards larger party size estimations. To limit the effect of differences in habitat structure and visibility between Taï and LuiKotale, the party compositions of the focal were recorded in a cumulative way by recording all individuals seen within 30 min and resetting the party composition at the start of each 30 min observation period. The sexual swellings of all females seen over the course of a day were scored on a range from 1–4 based on their degree of tumescence, with 4 scoring maximal tumescence. During the study period, all females in the Bompusa community at LuiKotale have been observed at least once to exhibit maximal tumescence. In the Taï East community, 7 of the 8 focalled females exhibited maximal tumescence at least once during the study period and in the Taï South community, 13 out of the 14 females exhibited maximal tumescence. CGB and the six field assistants collected all behavioural, party composition data using an identical protocol. CGB got trained in data collection on chimpanzees by LS and on bonobos by MS. CGB and LS trained all the research assistants. All data collectors used the same interface of the software Cybertracker on an android smartphone and we conducted inter-observer reliability tests between all observers and LS and MS to ensure consistency in data collection (IOR >90% across all observers).

**Statistics and reproducibility**. We used a series of General Linear Mixed Models (GLMMs) to test our predictions regarding species differences in patterns of female associations (for an overview of models see Tables 1 and 3). In the models, we used community ID instead of species as test variable as it allows us to assess whether or not the patterns were consistent across the two chimpanzee communities and thereby establish which results are species-specific and which are community-specific, although we cannot rule out population-level differences, which are thought to occur at least in chimpanzees[61]. In all models, we used each half-day focal follow as a data point and summarized the different parameters over this half-day period. As group sizes might vary systematically over the course of the day, we controlled in these models for the time of the day (three levels: morning, afternoon, and morning + afternoon if the focal started in the middle of the morning when the apes were lost the day before) by including it as a fixed effect into the models. To avoid pseudo replication, we added focal ID as a random factor and included random slopes for the time of the day within focal ID.

In the first two sets of models, we aimed at describing the overall patterns of species differences in female gregariousness without controlling for the potential mechanisms (i.e. without including food abundance or sexual signaling as a predictor in the analysis). This was done to first assess whether, numerically, Bompusa bonobo females are more gregariousness amongst females than Taï chimpanzee females.

In the first model, we tested for community differences in the numbers of female party members (P1). We used GLMMs with the rounded average number of female party members across the recorded 30 min party scans during a given focal follow as the response (Poisson error distribution). We included the community ID as the tested predictor (Model 1).

In the second and third models, we tested for community differences in the time females spent alone to test for prediction P2, P3b, and P4. We used GLMMs with the

percentage of time spent alone (Prediction P3b) during a focal follow as the response variable (beta error structure, as proportions are bound between 0 and 1). We included the community ID as a test predictor (Model 2a). In order to test for differences in female gregariousness independent of intersexual associations (Prediction P4), we rerun the same model with the percentage of time spent alone in relation to the time a given female was not associated with males as a response (Model 2b).

In the second set of models, we tested for the mechanisms underlying female gregariousness pattern in the two species by investigating the link between the presence of maximally tumescent females and female grouping patterns while controlling for variation in fruit availability and female dominance rank (Predictions P3a, P5, P6, P8, and P9).

We included the presence of maximally tumescent females in two different ways, avoiding a continuous count of swollen females, because of potential confound with the number of females as a response variable. Firstly, we incorporated it as a categorical variable with 3 levels to test Predictions P1a, P3, P4, P6, and P7: 0 (no maximally tumescent female [including focal swelling] individual present, to specifically test for Prediction P1a and P7), ≤1 (average of 1 or less but not 0 maximally tumescent females [including focal swelling] present during a given focal follow), and >1 (average of more than 1 maximally tumescent female [including focal swelling] present during the focal follow; Model 3a). The maximal number of maximally tumescent females present in the party was 8 in bonobos and 3 in both chimpanzee communities. Alternatively, in a different model, we included the percentage of time during which a maximally tumescent female was present in the party of the focal individual (including the focal individual swelling itself) as a further test of predictions P5, P6, P8, and P9 (Model 3b). For both models, we included the presence of maximally tumescent females in a two-way interaction with community as a test predictor, allowing to test for species differences in the effects of maximally tumescent females on female-female association patterns (Prediction P8).

Phenological data were not collected consistently and continuously in both study sites during our study period, preventing to use of phenological measures to assess fruit availability. However, we still controlled in models 3a and 3b for variation in food availability for each community in the following ways. We decided in a first step to use a behavioural proxy for food abundance, the monthly percentage of observation time spent feeding averaged across all focal individuals observed that month. Several studies in African apes indicate that when high-quality food is available in abundance (which corresponds to periods of large fruit abundance), less time is spent for feeding[71,98]. Furthermore, Taï chimpanzees were found to feed less long when fruit availability was high[72]. Since there was no published data on the relationship between feeding time and fruit availability in bonobos, we "validated" our behavioural proxy for bonobos at another study site where detailed information on fruit availability from phenological transects and activity budgets were available (see Supplementary Fig. 1). Accordingly, we used the average percentage of the time spent feeding by all focals of each community during a given month as a proxy for food availability and included it in an interaction with the community (as the relationship between feeding time and fruit abundance is likely species-specific) as a control variable into our models 3a and 3b. As an additional test to validate the robustness of our results of the influence of sexual swellings, we compared the directionality of the effect of our predictor variable with the ones derived from a model based on a reduced dataset (including only 13 months of bonobo data) with scores of monthly food availabilities at both sites (see Supplementary Table 2). In that reduced dataset the fruit availability for bonobos was calculated based on the numbers of all fruiting trees along the phenology transects in the home range of the study group and standardized from 0–1. This information was available for about two-thirds of the study days ($N = 137$). The food availability for chimpanzees was calculated on the fruiting patterns of chimpanzee food species along transects, extrapolated onto the home range using floristic plot data (for method see[46]) and standardized from 0–1 to make it comparable to the bonobo dataset.

We controlled in models 3a and 3b for potential effects of female dominance ranks by including them as a control predictor. We calculated dominance ranks in each of the study communities using a version of the Elo-rating method[99] (modified by Foerster et al.[100]). For the two chimpanzee communities, we used unidirectional submissive pant-grunt vocalizations as dominance interactions and for the bonobos we used the outcome of dyadic aggressive interactions with a clear winner and a clear loser (for further details see[101]).

All analyses were conducted in R 3.6.1[102] using the functions glmer for models with Poisson error distribution and glmmTMB for the models with delta error distribution, from the packages "lme4" and "glmmTMB" respectively[103,104]. In each model, we tested for the overall significance of the test predictors by comparing the full model to a null model comprising only the control predictors and random effects and slopes using a likelihood ratio test (LRT[105]). In the case that we rejected the null hypothesis, we assessed the significance of each single predictor variables by using an LRT between the full model and a reduced model comprising all the variables except the one to evaluate, using the drop1 function. If the LRT revealed that interaction had a p-value>0.1 (the threshold for a trend) we rerun the model without this interaction and reassessed the significance of all the predictors. The reference level for the community was set in our analyses to Bompusa (i.e. the bonobo community) allowing us to assess differences between the two chimpanzee communities and the bonobo one. When the LRT revealed a significant overall effect of community, we assessed individual p-values for differences across each of the community combinations by releveling our dataset

with one of the two chimpanzee communities being the reference level. This allowed us to get all the cross-comparisons and assess whether the community difference reflect species differences (i.e. both chimpanzee communities have the same pattern and differ from the bonobo community) or are community-specific (i.e. the two chimpanzee communities differ from each other).

For each model, we tested for collinearity issues between our predictor variables using the function vif from the package "car"[106]. Collinearity was not an issue (all vif<1.5). We also assessed model stability for each model by removing one level of the random effect at a time and recalculating the estimates of the different predictors, which revealed that the results were stable. Finally, we tested for over-dispersion in the models with Poisson and negative binominal error structure, which was not an issue in any of our models (all dispersion parameters <1.6)[107].

**Reporting summary**. Further information on research design is available in the Nature Research Reporting Summary linked to this article.

## Data availability
The datasets can be accessed under https://doi.org/10.5061/dryad.3r2280gh2.

## Code availability
The codes for the analysis in the current study are available from the corresponding author on request.

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

## Acknowledgements

We thank the Ministère de l'Enseignement Supérieur et de la Recherche Scientifique and the Ministère des Eaux et Forêts in Côte d'Ivoire, and the Office Ivoirien des Parcs et Réserves for permitting the study in Taï. We are grateful to the Centre Suisse de Recherches Scientifiques en Côte d'Ivoire. We thank the Institut Congolais pour la Conservation de la Nature (ICCN) for granting permission to work at LuiKotale and Lompole village for granting access to their forest. Many thanks go to Pascalle Dekker, Guilhem Duvot, Adeelia Goffe, Charlotte Grund, Luis Fernandez, Elodie Jocteur, Mathilde Grampp and Joanna Riera for invaluable assistance in collecting behavioural data in the field and to Sean Lee, Liza Moscovice, Tatiana Thomas, Patrick Tkaczynski and all the staff members of the LuiKotale Bonobo and Taï Chimpanzee Projects for their help in the field. We thank Frans de Waal and four anonymous referees for their constructive comments on the manuscript. This study was funded by the Max Planck Society and the European Research Council (ERC) under the European Union's Horizon 2020 research and innovation program awarded to C.C. (grant agreement no. 679787). Core funding for the Taï Chimpanzee Project has been provided by the Max Planck Society since 1997. Core funding for LuiKotale came from the Royal Zoological Society of Antwerp (KMDA) and private donors.

## Author contributions

M.S., C.G.B., C.B., C.C., R.W., G.H. conceptualized the study; M.S. and L.S. helped setting up the behavioural data collection; C.G.B. and a team of research assistants collected the behavioural data; C.G.B. and M.S. conducted the analysis; M.S. took the lead in writing the paper in consultation with C.G.B.; C.B., C.C., L.S., R.W., G.H. provided critical feedback and helped shape the analysis and writing of the paper; C.C., B.F., R.W., G.H., administer the fieldsites.

## Funding

## Competing interests

The authors declare no competing interests.
