## [Peer Review File · Communications Biology]

Attractiveness of female sexual signaling predicts differences in female grouping patterns between bonobos and chimpanzeesReviewers' comments:

Reviewer #1 (Remarks to the Author):

The submitted paper examines differences in sociality between bonobos and chimpanzees, sister species that exhibit stark divergence in some aspects of social behaviour, particularly female sociality. Here, the authors use comparative data to examine whether differences in female sexual signalling drive species differences in female party sizes. The study has some promising features. First, whereas comparative data on party sizes have a lot of confounds, these data were collected using identical methods by the same team of observers. Second, the communities being compared were of similar size and composition.

While these results are interesting, the context and interpretation of the study currently have many logical flaws that create misleading interpretations. My review is long in the hope that the explanations will be maximally useful for the authors in recrafting their report, but here are the key concerns: (1) The hypothesis testing framework is not appropriate to what the authors are testing and appears to have been forced onto the data post hoc; (2) The analysis is not well tailored to the predictions; (3) The results are erroneously interpreted as a challenge to a hypothesis that is not directly tested.

--- --- ---

The problem that leads the manuscript astray is that the introduction introduces a confusing theoretical framework that pits two hypotheses, the socioecology and female sexuality hypotheses, against one another. However, these ideas attempt to explain different things. The socioecological hypothesis explains an increase in female gregariousness, and by extension posits that this could afford females the opportunity to develop stronger social bonds. The female sexuality hypothesis, as described in the ms, explains reduced male sexual aggression and only incidentally might affect female associations with other females. The explanations given for the hypotheses actually encapsulate several different hypotheses which could be independently supported or falsified. For example, ecological factors could predict female gregariousness without that necessarily leading to any of the other effects cited (e.g., female dominance, bonds, lack of coercion), and vice versa, increased gregariousness could be caused by other factors and still yield those effects. While the authors acknowledge that the hypotheses are not mutually exclusive, the setup of the predictions implies that they are weighed against each other. This is particularly strange as the paper does not test any direct predictions of the socioecology hypothesis with data on food availability or distribution, and in fact, the authors still conclude that ecological factors likely underlie these patterns.

Neither hypothesis is well explained. In stating the conclusion, the authors note that their data lead them to reject the argument that female bonobos are motivated to be together by virtue of their ecology alone. This IS an appropriate interpretation of these results and is an important finding. But, that was never a prediction of the socioecology hypothesis. The hypothesis argues that features of feeding ecology, particularly a reduced spatial and temporal clumping of food reduces the constraints of contest feeding competition for bonobos females. Notably (a) this hypothesis predicts that bonobo females will join larger parties generally, not just that they will join other females (though this is the net outcome); (b) there is not a prediction that features of the ecology should produce a stronger intrinsic motivation for females to associate with other females; (c) because the hypothesis is agnostic as to why females might join larger parties, the finding that Bompusa bonobo females ARE more gregarious is consistent, even if that can be explained by other proximate factors.

It is unclear how the authors apply the 'female sexuality' hypothesis to their predictions. The simplest interpretation of this argument would be that females are found together incidentally due to their mutual attraction to/from males when swollen. This is not so clearly stated, and the analysis is not set up explicitly to test this since it looks at association with females independently of association with males and analyses the swelling status of the other females in the party rather than the focal. Also, if true, this hypothesis does not leave much room for female bonding unless feeding competition were also relaxed. The authors include some discussion that females might be themselves attracted to swollen females, but this generates different predictions, including that bonobos should exhibit larger female party sizes when swollen females are present.

The predictions about male gregariousness did not logically follow from the hypotheses and ignored any other influences on male sociality (like the importance of collective male territorial

defence in chimpanzees). Prediction³ of the sexuality hypothesis is particularly illogical as it posits an effect independent of patterns of sexual swellings even though the patterns of sexual swellings are meant to be the very basis for the prediction.

The analysis is not well tailored to the questions. The authors elected to use focals as their unit of analysis, both because focal follows minimise observation bias and because this structure could accommodate individual differences in sampling and in gregariousness. This is good. It is less clear why male focal follows were used at all (as above), and it is particularly strange to use male and female follows in the same model. There were separate predictions about males and females, and this structure forces the use of awkward interactions between community and sex to test the central prediction that should be tested based on female behaviour alone. I assume the authors wanted to represent as many observed parties, which is at odds with the justification for using the focals as the unit of analysis in the first place. In the SI, male and female-only models are reported, which seems preferable.

Given that the influence of sexually swollen females is central to the prediction, the treatment of this issue was inadequate. If the analytical structure is based around focal individuals, why was the focal female's swelling status considered not instead of (or in addition to) the number of swollen females in the party. It is more logical to assume that a female may be attracted to a large party because she is swollen than because other females are.

A continuous count of swollen females was used, creating a confound with number of females as the outcome variable. The simulation exercise illustrates that it is still mathematically possible to detect the alternative outcome, but I was not convinced that this represented the nature of real data. If this is the prediction that the authors hang their hats on, it is important that the model itself directly support or refute it. There should be more robust ways of dealing with this issue. Given a fairly small community size, there will not be a linear relationship between the number of swollen females and party size, and the party size should reach an asymptote beyond which adding more swollen females cannot yield a greater increase. Using a factor of 0, 1, or more than 1 swollen females likely captures most of the variation and produces less of a confound.

Also, if the focal is the unit of analysis, the authors should consider including controls for age and rank, or at least discuss whether the age breakdowns of their communities are approximately similar given previously documented differences between nulliparous and parous females in gregariousness and sexual signalling.

Figure 1A is striking. While mean party sizes are only a little different, there is a clear tendency of female bonobos to be in very large parties. Chimpanzee parties rarely contained more than 7 females while many bonobos parties included 8-15 females. It does not make sense that this difference disappeared when the number of possible females was considered because the number of female bonobos (13) was equivalent to the number of female chimpanzees (11 in one group, 15 in the other). I am concerned that the analysis does not deal with the differences in distribution between the bonobo and chimpanzee party sizes and MTF. Given that there are minimal size differences between groups, it is strange to consider % of available females if the actual number is more functionally relevant to feeding competition and ability to affiliate.

The discussion raises interesting issues regarding this comparison, notably that the female Tai chimpanzees have previously been found to be more gregarious than those in other chimpanzee populations for reasons that are unusual (predation). Please address why that matters more directly, i.e., that this data suggests more overlap in female gregariousness for these species than previously recognised but only when considering an extreme for chimpanzees. Also, while prior data on bonobos were discussed based on methodological grounds, one could present more of the actual comparative figures to evaluate where *Bompusa* may lie on the range of bonobo variation. Another notable peculiarity that is not discussed is that Tai chimpanzees have been reported by Deschner & Boesch and others to exhibit much longer, bonobo-like periods of sexual cycling within the interbirth interval compared with other chimpanzee populations. This somewhat undermines the premise of the comparison. The data did indicate a higher occurrence of swollen females in the bonobo group, though it is unclear how much that is driven by the fact that more females were observed. Since this difference is critical to the objectives of the study, the authors should better set up this comparison by outlining the available data that the species differ systematically with respect to cycling and situating the study populations within it. This lit has been confusing for bonobos. Early reports had it that bonobos cycle for longer but also have more swelling days in the menstrual cycle, but the report of this population (Douglas et al. 2016) has bonobo females swelling for 25% of the cycle, similar to or less than chimpanzees.

In sum, even though the *Bompusa* bonobos and Tai chimpanzees were more similar to one

another than most chimpanzee-bonobo comparisons, the authors found that bonobo parties contained on average more females (but not males) than chimpanzee parties, and bonobo females were often found in aggregations considerably larger than were ever observed in the chimpanzees. While the data indicate that sexual behaviour incentivises these associations, it does not – and cannot – rule out the underlying influence of species differences in feeding ecology.

Reviewer #2 (Remarks to the Author):

Martin Surbeck Hohmann: Extended periods of sexual signaling explains differences in grouping patterns between bonobos and chimpanzees

5066 Nature Research

This is a much-needed evaluation of the grouping patterns of wild chimpanzees and bonobos. The traditional story on the difference between both species relates to more female social cohesion in bonobos, caused by differences in ecology, but this study questions this explanation. The new element here is the use of an identical methodology for populations of both species, one that does not bias towards large and noisy parties, but relies on focal (individual) follows. The result is a more objective comparison, and the conclusion that the female grouping patterns differ little, especially if female sexual cycles are taken into account. I found this most enlightening. It offers much food for thought.

I have a few general remarks, and must say that I am no specialist in glmm and glmer statistical analyses, so am unable to judge details of this part of the manuscript.

The term "codominance" bothers me, because it has no accepted definition. In fact, it is rare or absent in the animal behavior literature. We have "ambiguous" or "undetermined" dominance, we also have "egalitarianism," and the absence of dominance, but codominance is truly a nonexistent concept. Perhaps it was invented by Kano in the days that female dominance in bonobos was a controversial topic. Looking at Table 3 in Surbeck & Hohmann (2013) there is no reason to use this term, because the table says that in the bonobos of LuiKotale the first six (!) rank positions in the social hierarchy are occupied by females. This is female dominance, plain and simple. In fact, I have never heard of a bonobo group, captive or wild, in which the alpha individual is a male. I'd suggest to either drop the term "codominance" entirely or give a justification of why you'd use it and also how you define it. It literally would mean that both males and females are dominant, which conflicts with the classical dominance concept that assumes asymmetry.

Second, I would emphasize even more than you do in the Discussion how the literature on chimpanzees has been shaped by observations of Eastern populations. For example, the chimpanzees at Tai do not seem to fit the "high rates of coercive aggression (Muller et al., 2009)" generalization if I understand the writings of Boesch on Tai (e.g. Stumpf & Boesch, 2010). Most of the emphasis in chimpanzees on sexual coercion, violent warfare, and female dispersal seems to come from Gombe, Mahale, Kibale, and so on. There is nothing wrong with this, of course, but the Tai chimpanzees have always seemed different.

You explain that Tai chimpanzees may have more bonobo like grouping patterns and that the differences with bonobos in behavior may be less stark. Instead of trying to come up with a generalized picture of chimpanzee social life, therefore, I would emphasize the within-chimpanzee variability, which actually makes this whole comparison so interesting. Because even though the female data on both species seem similar in many ways, the male data are not at all. According to your study, Tai chimpanzees and bonobos differed especially in how males spend their time. Table 2: Bonobo males are almost never alone, whereas chimpanzee males are often alone and more often in all male parties.

It seems that the main difference is that bonobos stay around their mother (line 400: "among bonobo females, mothers with adult sons seemed to spend the least time alone"), whereas chimpanzee males don't, and spend more time with each other. You refer to ideas that this

difference in alliances may be due to the larger number of tumescent females in bonobo groups. This idea was first formulated in my book *Bonobo: The Forgotten Ape* (p. 140). "... male alliances in other primates are mostly instruments to keep competitors away from a highly contested female, the reason for such cooperation is eliminated if multiple females are sexually attractive at once."

Line 143 "The two hypotheses are not mutually exclusive, as ecological factors likely affect overt sexual signaling by females ..." You never return to this idea. It is intriguing because one would still expect that ecological factors play a role. After all, your paper is mostly concerned with proximate factors in social grouping. We still need to know why they are different between both Pan species.

Study material: The bonobo data come from only one community, which is a limitation of the study. I know this cannot be changed, but it will be good to stress that this is not a comparison between chimpanzees and bonobos in general, but between two specific chimpanzee communities at one location and a single bonobo community.

Line 551 "... an increased general gregariousness among females might not be the underlying species difference ..." Perhaps, but didn't Furuichi emphasize that bonobo females call each other at night and build their nests not far from each other? Is this also true for chimpanzee females at Tai? It has always struck me as a sign that bonobo females rely more on each other.

Despite these comments and questions, thanks for a thought-provoking paper.

Frans de Waal

Reviewer #3 (Remarks to the Author):

This paper examines species differences in female and male gregariousness in chimpanzees and bonobos. Prior research suggests that female bonobos are more gregarious than female chimpanzees. One hypothesis links this to differences in the feeding ecology of the two species. A more abundant and less fluctuating food supply for bonobos has been argued to permit females in this species to gather together more frequently than female chimpanzees. Alternatively, others have proposed that bonobo females experience extended periods of estrus compared to female chimpanzees. This, in turn, attracts many male bonobos to them, creating large subgroups or parties in the process. Female bonobos, through an unspecified mechanism (see lines 135 – 142 this manuscript), are also attracted to these mixed-sex parties containing other estrous females, leading to increased gregariousness among female bonobos.

The authors ostensibly test these two hypotheses, which they admit are not mutually exclusive, by collecting comparative data on the grouping behavior of bonobos and chimpanzees. They make several predictions based on what they call the "female ecology hypothesis" (lines 193 – 201) and "female sexuality hypothesis" (lines 208 – 217). Their findings indicate that:

- 1) Female bonobos are more gregarious, as measured by female party size, than female chimpanzees. This difference, however, vanishes when controlling for the number of females in bonobo and chimpanzee groups and the presence of estrus bonobo and chimpanzee females.
- 2) The percentage time spent alone by female bonobos and chimpanzees did not differ nor did the amount of time females in each species spent with each other.

One other hidden result seems to indicate that:

- 3) Bonobo females are in estrus more frequently than are chimpanzee females. This is suggested by the findings that maximally tumescent bonobo females were present much more often than maximally tumescent chimpanzee females and that the average number of the former in parties was three times higher than the number of the latter. A direct tally of how often females in each species spent in estrus is not provided, however.

From these three findings, the authors conclude that extended sexual signaling accounts for species differences in grouping patterns between bonobos and chimpanzees. They go further, though, and make a bold claim that these results also explain why female bonobos are socially dominant to male conspecifics and why the latter do not sexually coerce the former.

This is an interesting paper, but there are several problems. These problems involve the paper's formulation, analysis, and interpretations. First, the paper is set up as a test of two competing hypotheses, essentially whether food or sex, influence bonobo and chimpanzee grouping patterns. At the outset we need data on the food supply for bonobos and chimpanzees to adequately test the female ecology hypothesis. None are provided, however. Alternatively, to test the female sexuality hypothesis, we require information on how often female bonobos and chimpanzees spend in estrus. As noted above, we are not furnished direct information on this matter. An additional problem is that, as the authors themselves admit (lines 145 – 146), the two hypotheses are not mutually exclusive, and the factors that influence the feeding ecology hypothesis might ultimately affect the reasons why extended sexual signaling in bonobos might exist. A more illuminating paper would have made an attempt to disentangle these, and with that, provide a stronger test of the two hypotheses.

Analytically, data on the grouping behavior of bonobos and chimpanzees is used to test the two hypotheses. But as indicated above, grouping data alone is insufficient to test them. One of the main results that female bonobos are more gregarious than female chimpanzees appears to be consistent with the female ecology hypothesis, but this is tempered by the finding that this difference vanishes when one controls for the presence of estrus females. The significance of this result is questionable, as the authors themselves admit, their analysis uses a subset of the dependent variable as an independent variable. Another important finding that the difference in female grouping patterns disappears when one controls for the number of females in each species is interesting, but without information about the food supply, one can't evaluate it. Similar feeding conditions for the two species over the relatively short duration of study, a possibility consistent with the feeding ecology hypothesis, might explain the lack of difference.

Finally, the authors go far beyond the data they present and argue that their findings indicate why female bonobos are socially dominant to male conspecifics and why the latter do not sexually coerce the former as male chimpanzees do to their female conspecifics. No data are provided on either of these points, however. The authors note (lines 549-552) that we require more information on male aggression toward females, female-male dominance relationships, and female sexuality to make such conclusions. None are provided. And in the next sentence, they admit that the evolutionary origin of female cooperation in bonobos that leads to their ability to dominate males remains "puzzling."

In sum, this paper contains some interesting comparative data regarding the grouping behavior of bonobos and chimpanzees. What to make of these data, though, is difficult to discern.

Response to referees

We thank all the reviewers for their efforts to provide helpful and thorough feedback on our manuscript. These comments allowed us to significantly improve the clarity and the focus of our manuscript. Below we provide detailed reply to each of the comments made by the reviewers and explain how we addressed them in the revised version of the manuscript in bold.

Reviewer #1 (Remarks to the Author):

The submitted paper examines differences in sociality between bonobos and chimpanzees, sister species that exhibit stark divergence in some aspects of social behaviour, particularly female sociality. Here, the authors use comparative data to examine whether differences in female sexual signalling drive species differences in female party sizes. The study has some promising features. First, whereas comparative data on party sizes have a lot of confounds, these data were collected using identical methods by the same team of observers. Second, the communities being compared were of similar size and composition.

We thank the reviewer for this overall positive evaluation of our study.

While these results are interesting, the context and interpretation of the study currently have many logical flaws that create misleading interpretations. My review is long in the hope that the explanations will be maximally useful for the authors in recrafting their report, but here are the key concerns: (1) The hypothesis testing framework is not appropriate to what the authors are testing and appears to have been forced onto the data post hoc; (2) The analysis is not well tailored to the predictions; (3) The results are erroneously interpreted as a challenge to a hypothesis that is not directly tested.

The revised version of our manuscript has a more focused hypotheses framework. We now test exclusively for proposed species differences and underlying processes in female grouping patterns, as these differences were hypothesized to translate in major behavioural differences. The revised statistical models are more tailored to test the predictions since they focus on female focal follows and investigate now only female-female gregariousness and not male gregariousness. They furthermore control for fluctuations in food availability to rule out the possibility that female sexual signals are only expressed during periods with specific ecological conditions and to rule out that the observed female-female grouping patterns result from fluctuation in food abundance rather than as a response to female sexual swellings themselves.

The problem that leads the manuscript astray is that the introduction introduces a confusing theoretical framework that pits two hypotheses, the socioecology and female sexuality hypotheses, against one another. However, these ideas attempt to explain different things. The socioecological hypothesis explains an increase in female gregariousness, and by extension posits that this could afford females the opportunity to develop stronger social bonds. The female sexuality hypothesis, as described in the ms, explains reduced male sexual aggression and only incidentally might affect female associations with other females.

The explanations given for the hypotheses actually encapsulate several different hypotheses which could be independently supported or falsified. For example, ecological factors could predict female gregariousness without that necessarily leading to any of the other effects cited (e.g., female

dominance, bonds, lack of coercion), and vice versa, increased gregariousness could be caused by other factors and still yield those effects. While the authors acknowledge that the hypotheses are not mutually exclusive, the setup of the predictions implies that they are weighed against each other. This is particularly strange as the paper does not test any direct predictions of the socioecology hypothesis with data on food availability or distribution, and in fact, the authors still conclude that ecological factors likely underlie these patterns.

We rewrote the introduction of the theoretical framework in a way that doesn't oppose these hypotheses and focusses on different processes explaining variation in female-female gregariousness. Female-female gregariousness has been proposed to be the underlying social difference leading to many species' differences on a behavioural level. The hypothesized processes are that bonobo female are motivated to be together by virtue of the ecology alone or that patterns of sexual swellings are closely linked to female gregariousness. We largely removed the parts that link sexuality with male aggression from the introduction and only discuss them as implications later in the manuscript. More crucially, as indicated by the reviewers, we missed the incorporation of ecological measures in our models to evaluate the impact of factors such as food abundance on grouping patterns. In order to show that the observed pattern were not solely driven by ecological factors and tease apart the role of ecology and of female sexuality, we incorporated now in all our analysis a measure of feeding time as a proxy for food abundance. In a recently published paper on the same chimpanzee population as our study population, we found, combining detailed phenological and behavioural observations, that chimpanzee feed less long when food availability was higher (Valé et al., 2020). As for the bonobos, since we did not collect detailed phenological data (only number of fruiting trees along transects for a subset of our observation months), we compared phenological and feeding data at another of our study site, Kokolopori. Kokolopori presents similar ecological conditions as LuiKotale and is within 450 km. We found in this population that, as for the chimpanzees in Taï, bonobo fed for shorter duration when food availability was high. Overall, feeding time is thus a reliable indicator of food abundance in both populations and we used this measure as a proxy for food abundance. Please note that feeding time is negatively correlated to food abundance biomass and fruit consumption (which correspond to periods of high fruit availability) in other african great ape populations (Knott, 2005; Masi et al., 2009; Watts, 1988). The behavioural proxy we use here (feeding time) is furthermore widely applicable across primates (e.g. Hanya, 2004; Harris et al., 2010). As an additional test for the robustness of our results, we run the models on a subset of the data including the months with number of fruit trees in LuiKotale and compared the model estimates to the larger model with percentage feeding time. The results were similar so that we are confident that our results are not driven by the use of a proxy for food availability.

Neither hypothesis is well explained. In stating the conclusion, the authors note that their data lead them to reject the argument that female bonobos are motivated to be together by virtue of their ecology alone. This IS an appropriate interpretation of these results and is an important finding.

As explained in detailed above, in the revised version of the manuscript we incorporated a behavioural proxy for food abundance allowing us to statistically rule out that bonobo females are more gregarious than chimpanzee females by virtue of ecology alone. This remains thus an important finding of the revised version of the manuscript but we know have more empirical basis to discuss it.

But, that was never a prediction of the socioecology hypothesis. The hypothesis argues that features of feeding ecology, particularly a reduced spatial and temporal clumping of food reduces the constraints of contest feeding competition for bonobos females. Notably (a) this hypothesis predicts that bonobo

females will join larger parties generally, not just that they will join other females (though this is the net outcome);

We agree with the reviewer. However, since the revised version of our manuscript focus now on female-female gregariousness in order to streamline and clarify our predictions, our analysis now target female-female associations alone and we only discuss implications for male association patterns in the discussion but do not test these patterns anymore.

(b) there is not a prediction that features of the ecology should produce a stronger intrinsic motivation for females to associate with other females;

This is correct and is no longer an argument made in the revised version of the manuscript.

(c) because the hypothesis is agnostic as to why females might join larger parties, the finding that Bompusa bonobo females ARE more gregarious is consistent, even if that can be explained by other proximate factors.

This is correct and in the revised version we have a first set of models testing for the proposed differences in female gregariousness which are agnostic to the underlying processes and just test for the validity of the basic statement of higher female gregariousness in bonobos. It is only the second sets of models that test for species differences in the underlying processes influencing variation in female gregariousness. To account for the valid objections by the reviewer, we reformulated the whole hypothesis and prediction section of the manuscript.

It is unclear how the authors apply the 'female sexuality' hypothesis to their predictions. The simplest interpretation of this argument would be that females are found together incidentally due to their mutual attraction to/from males when swollen. This is not so clearly stated, and the analysis is not set up explicitly to test this since it looks at association with females independently of association with males and analyses the swelling status of the other females in the party rather than the focal.

In the revised version we now elaborate more on potentially different mechanisms of female attraction to other maximally tumescent females, including mutual attraction to males, as stated by the reviewer. We highlight how inter-female variation in gregariousness can potentially inform about potential underlying mechanisms.

While the amount of focal follows with maximally tumescent did not allow for a restriction of the analysis to only the influence of the swelling stage of the focal female, we present this data quantitatively in the revised version of the manuscript.

To account for potential confound between the number of females and the number of maximally tumescent females in the party we included in the revised analysis the presence of MTF as a categorical variable with 3 levels (no MTF present, average of 1 or less MTF present, and more than 1 MTF present). We also repeated our analysis replacing this categorical factor by a quantitative one: the percentage time spent with at least one MTF in the party. The results were consistent across the two analyses and showed that, even after controlling for variation in food abundance (using our behavioural proxy) the effect of the presence of MTF (either categorical or as a percentage) was associated with a significantly stronger increase in the number of female in the party in bonobos as compared to chimpanzees.

Also, if true, this hypothesis does not leave much room for female bonding unless feeding competition were also relaxed. The authors include some discussion that females might be themselves attracted to swollen females, but this generates different predictions, including that bonobos should exhibit larger female party sizes when swollen females are present.

According to the reviewer's comments, we modified the prediction of the hypothesis and actually find exactly what the reviewer predicted, namely that the presence of MTF leads to a larger increase in party sizes in bonobos as compared to chimpanzees (see above).

The predictions about male gregariousness did not logically follow from the hypotheses and ignored any other influences on male sociality (like the importance of collective male territorial defence in chimpanzees).

In order to streamline the arguments and account for the reviewers comment below, we restricted the analysis in the revised version of the manuscript to female-female gregariousness and therefore omitted aspects of male gregariousness. We believe that this led to a much more focused and clearer manuscript with a clear take-home message.

Prediction3 of the sexuality hypothesis is particularly illogical as it posits an effect independent of patterns of sexual swellings even though the patterns of sexual swellings are meant to be the very basis for the prediction.

We agree with the reviewer and now omitted this prediction from the revised version of the manuscript.

The analysis is not well tailored to the questions. The authors elected to use focals as their unit of analysis, both because focal follows minimise observation bias and because this structure could accommodate individual differences in sampling and in gregariousness. This is good. It is less clear why male focal follows were used at all (as above), and it is particularly strange to use male and female follows in the same model.

This is a valid point and we thank the reviewer for this comment. In order to address this issue we restricted the analysis only to female focal follows in the revised version of the manuscript and focus on female-female associations, as they are the crucial aspect that we wanted to investigate.

There were separate predictions about males and females, and this structure forces the use of awkward interactions between community and sex to test the central prediction that should be tested based on female behaviour alone. I assume the authors wanted to represent as many observed parties, which is at odds with the justification for using the focals as the unit of analysis in the first place. In the SI, male and female-only models are reported, which seems preferable.

We agree that this was a drawback of the previous version of the manuscript. As mentioned above, in order to streamline the presentation of the hypothesis and the discussion and improve the clarity of the manuscript, we restricted the analysis to the female association patterns using female focal follows only.

Given that the influence of sexually swollen females is central to the prediction, the treatment of this issue was inadequate. If the analytical structure is based around focal individuals, why was the focal

female's swelling status considered not instead of (or in addition to) the number of swollen females in the party. It is more logical to assume that a female may be attracted to a large party because she is swollen than because other females are.

This is a good remark but as mentioned above we did not have enough days when the focal female was maximally tumescent (especially in the chimpanzees) to run a meaningful analysis on that factor alone. However, , we have modified the way we integrated swelling status of females into our analyses. We now made the number of sexually swollen females in the party categorical (0, 1 or more than 1) to avoid confounding the count of females in the party with the count of maximally tumescent females in the party, which may covary just mathematically. We agree that larger party around swollen females can result from either swollen females being attracted to larger parties, or swollen females attracting themselves more other females or a combination of the two mechanisms. We do not think that focusing solely on the swelling status of the focal females would allow disentangling these two hypotheses more than our current approach. In fact, if a focal swollen female has more females around her in the party it could still be because she actively sought larger parties to associate with or because she attracted more other females towards her party. Please note that for our study the specific mechanism is not crucial to the theory tested since they both lead to larger party because of extended periods of maximal tumescence in bonobo as compared to chimpanzees. In turn, our conclusion that female sexual signalling is the main proximate driver of the difference found between the two species, even after controlling for ecological confound as we do here, does not rely on one specific mechanism.

A continuous count of swollen females was used, creating a confound with number of females as the outcome variable. The simulation exercise illustrates that it is still mathematically possible to detect the alternative outcome, but I was not convinced that this represented the nature of real data. If this is the prediction that the authors hang their hats on, it is important that the model itself directly support or refute it. There should be more robust ways of dealing with this issue. Given a fairly small community size, there will not be a linear relationship between the number of swollen females and party size, and the party size should reach an asymptote beyond which adding more swollen females cannot yield a greater increase. Using a factor of 0, 1, or more than 1 swollen females likely captures most of the variation and produces less of a confound.

We included the presence of maximally tumescent females (MTF) in the way suggested by the reviewer (0, one, and more than 1), as well as the percentage of time during a focal follow that at least one maximally tumescent female was present in the party. As mentioned above both analyses revealed the same results and confirmed that the presence of MTF had a stronger impact on female gregariousness in bonobos as compared to chimpanzees.

Also, if the focal is the unit of analysis, the authors should consider including controls for age and rank, or at least discuss whether the age breakdowns of their communities are approximately similar given previously documented differences between nulliparous and parous females in gregariousness and sexual signalling.

We followed the reviewer's suggestion and included the rank of the focal female in our analyses. Rank did not have a significant effect on female gregariousness. As for the age, we included only one nulliparous female as a focal individual and included this information in the revised version of the manuscript. Furthermore, we provide a table in the supplements with an overview of the focal

individuals and of their estimated age (since most focal individuals were born before the beginning of the long-term study in each study sites or before the group was fully habituated and the individuals fully identified). The age distribution was similar between the 3 study communities.

Figure 1A is striking. While mean party sizes are only a little different, there is a clear tendency of female bonobos to be in very large parties. Chimpanzee parties rarely contained more than 7 females while many bonobos parties included 8-15 females. It does not make sense that this difference disappeared when the number of possible females was considered because the number of female bonobos (13) was equivalent to the number of female chimpanzees (11 in one group, 15 in the other). I am concerned that the analysis does not deal with the differences in distribution between the bonobo and chimpanzee party sizes and MTF. Given that there are minimal size differences between groups, it is strange to consider % of available females if the actual number is more functionally relevant to feeding competition and ability to affiliate.

We thank the reviewer for pointing this out. In the revised version of the manuscript we now focus on the number of female in the party and the time female spent alone and do not present anymore the analysis on the percentage of females from the community present in the party. This streamlines the result of our paper and contributes to a more focused manuscript.

The discussion raises interesting issues regarding this comparison, notably that the female Tai chimpanzees have previously been found to be more gregarious than those in other chimpanzee populations for reasons that are unusual (predation). (Mulavwa et al., 2008) i.e., that this data suggests more overlap in female gregariousness for these species than previously recognised but only when considering an extreme for chimpanzees. Also, while prior data on bonobos were discussed based on methodological grounds, one could present more of the actual comparative figures to evaluate where Bompusa may lie on the range of bonobo variation.

We thank the reviewer for these inspirational ideas and added in the discussion the points raised here.

Given the differences in the methodology of data collection on female gregariousness, there are no actual comparable bonobo data published to relate the LuiKotale data to. However, we add the numbers derived from a different methodological approach in the revised version of the manuscript which indicate that our bonobo population has larger female party sizes (4.5) than Wamba (3.2) and Lomako (3.0). (Lines 591ff)

Another notable peculiarity that is not discussed is that Tai chimpanzees have been reported by Deschner & Boesch and others to exhibit much longer, bonobo-like periods of sexual cycling within the interbirth interval compared with other chimpanzee populations. This somewhat undermines the premise of the comparison. The data did indicate a higher occurrence of swollen females in the bonobo group, though it is unclear how much that is driven by the fact that more females were observed. Since this difference is critical to the objectives of the study, the authors should better set up this comparison by outlining the available data that the species differ systematically with respect to cycling and situating the study populations within it. This lit has been confusing for bonobos. Early reports had it that bonobos cycle for longer but also have more swelling days in the menstrual cycle, but the report of this population (Douglas et al. 2016) has bonobo females swelling for 25% of the cycle, similar to or less than chimpanzees.

In the revised version of the manuscript we provide information about the percentage of times when maximally tumescent females are in the parties, as well as how often the actual focal was maximal tumescent. Both measures indicate that the presence of a maximally tumescent females is more frequent in bonobos at LuiKotale than in chimpanzees at Tai, at least during our study period. (Line 423 ff).

In sum, even though the Bompusa bonobos and Tai chimpanzees were more similar to one another than most chimpanzee-bonobo comparisons, the authors found that bonobo parties contained on average more females (but not males) than chimpanzee parties, and bonobo females were often found in aggregations considerably larger than were ever observed in the chimpanzees. While the data indicate that sexual behaviour incentivises these associations, it does not – and cannot – rule out the underlying influence of species differences in feeding ecology.

We completely agree with the reviewer that our formal analysis did not allow us to rule out that feeding ecology drove the female-female association. By incorporating now a measure of food abundance into our analyses we can better control for the influence of ecology and are more confident in concluding that sexual signalling is the main factor that drives higher female-female gregariousness in bonobos and not ecology. We believe that our streamlined and more controlled analyses and fully revised hypotheses and conclusions provide a clearer message.

Reviewer #2 (Remarks to the Author):

Martin Surbeck Hohmann: Extended periods of sexual signaling explains differences in grouping patterns between bonobos and chimpanzees

5066 Nature Research

This is a much-needed evaluation of the grouping patterns of wild chimpanzees and bonobos. The traditional story on the difference between both species relates to more female social cohesion in bonobos, caused by differences in ecology, but this study questions this explanation. The new element here is the use of an identical methodology for populations of both species, one that does not bias towards large and noisy parties, but relies on focal (individual) follows. The result is a more objective comparison, and the conclusion that the female grouping patterns differ little, especially if female sexual cycles are taken into account. I found this most enlightening. It offers much food for thought.

We would like to thank the reviewer for this positive evaluation of the manuscript.

I have a few general remarks, and must say that I am no specialist in glmm and glmer statistical analyses, so am unable to judge details of this part of the manuscript.

The term “codominance” bothers me, because it has no accepted definition. In fact, it is rare or absent in the animal behavior literature. We have “ambiguous” or “undetermined” dominance, we also have “egalitarianism,” and the absence of dominance, but codominance is truly a nonexistent concept. Perhaps it was invented by Kano in the days that female dominance in bonobos was a controversial topic. Looking at Table 3 in Surbeck & Hohmann (2013) there is no reason to use this term, because the table says that in the bonobos of LuiKotale the first six (!) rank positions in the social hierarchy are occupied by females. This is female dominance, plain and simple. In fact, I have never heard of a bonobo

group, captive or wild, in which the alpha individual is a male. I'd suggest to either drop the term "codominance" entirely or give a justification of why you'd use it and also how you define it. It literally would mean that both males and females are dominant, which conflicts with the classical dominance concept that assumes asymmetry.

We agree that the term co-dominance can be misleading and is a non-concept. We removed this term and state in the revised version of the manuscript that many females outrank all adult males in the group.

Second, I would emphasize even more than you do in the Discussion how the literature on chimpanzees has been shaped by observations of Eastern populations. For example, the chimpanzees at Tai do not seem to fit the "high rates of coercive aggression (Muller et al., 2009)" generalization if I understand the writings of Boesch on Tai (e.g. Stumpf & Boesch, 2010). Most of the emphasis in chimpanzees on sexual coercion, violent warfare, and female dispersal seems to come from Gombe, Mahale, Kibale, and so on. There is nothing wrong with this, of course, but the Tai chimpanzees have always seemed different.

You explain that Tai chimpanzees may have more bonobo like grouping patterns and that the differences with bonobos in behavior may be less stark. Instead of trying to come up with a generalized picture of chimpanzee social life, therefore, I would emphasize the within-chimpanzee variability, which actually makes this whole comparison so interesting. Because even though the female data on both species seem similar in many ways, the male data are not at all. According to your study, Tai chimpanzees and bonobos differed especially in how males spend their time. Table 2: Bonobo males are almost never alone, whereas chimpanzee males are often alone and more often in all male parties.

Following reviewer 1 and in order to enhance the clarity and scope of our paper, we now focused on female sociality and controlled for ecological factors, we thus do not present the male results anymore. Our revised analysis reveals that, even after controlling for ecology, female sexual signalling affects female sociality differently in LuiKotale bonobos than in Tai chimpanzees, indicating the importance of this trait in setting the species apart. We nevertheless emphasize the variability observed in chimpanzees in the discussion of the revised version of the manuscript. (Line 572ff)

It seems that the main difference is that bonobos stay around their mother (line 400: "among bonobo females, mothers with adult sons seemed to spend the least time alone"), whereas chimpanzee males don't, and spend more time with each other. You refer to ideas that this difference in alliances may be due to the larger number of tumescent females in bonobo groups. This idea was first formulated in my book *Bonobo: The Forgotten Ape* (p. 140). "... male alliances in other primates are mostly instruments to keep competitors away from a highly contested female, the reason for such cooperation is eliminated if multiple females are sexually attractive at once."

We add this reference in the part in the revised discussion where we discuss the implications of the female bonobo sexuality on male behaviour (de Waal, 1997) (L707)

Line 143 "The two hypotheses are not mutually exclusive, as ecological factors likely affect overt sexual signaling by females ..." You never return to this idea. It is intriguing because one would still expect that ecological factors play a role. After all, your paper is mostly concerned with proximate factors in social grouping. We still need to know why they are different between both Pan species.

In the revised version of the manuscript we include a proxy for ecological factors (food availability, see reply to reviewer 1 comments for more details) and verify the robustness of our results with actual fruiting tree data. This allows us to present and discuss a more detailed picture of the interplay between sexuality and ecology.

Study material: The bonobo data come from only one community, which is a limitation of the study. I know this cannot be changed, but it will be good to stress that this is not a comparison between chimpanzees and bonobos in general, but between two specific chimpanzee communities at one location and a single bonobo community.

We emphasized this point stronger in the revised version of the manuscript. (L572ff)

Line 551 "... an increased general gregariousness among females might not be the underlying species difference ..." Perhaps, but didn't Furuichi emphasize that bonobo females call each other at night and build their nests not far from each other? Is this also true for chimpanzee females at Tai? It has always struck me as a sign that bonobo females rely more on each other.

It is true for bonobos that some of the females build their nests close to each other (also including some male nests especially for son often nesting next to their mother even while adults). In chimpanzees at Tai, females also tend to nest next to each other and higher up in the tree than males. Males in Tai nest more solitarily at some distance from the females and from other males (personal observation). This nesting pattern of females at Tai might be more constrained by leopard predation than in LuiKotale where leopard density is lower (unpublished camera trap data).

Despite these comments and questions, thanks for a thought-provoking paper.

Thank you for this positive judgement.

Frans de Waal

Reviewer #3 (Remarks to the Author):

This paper examines species differences in female and male gregariousness in chimpanzees and bonobos. Prior research suggests that female bonobos are more gregarious than female chimpanzees. One hypothesis links this to differences in the feeding ecology of the two species. A more abundant and less fluctuating food supply for bonobos has been argued to permit females in this species to gather together more frequently than female chimpanzees. Alternatively, others have proposed that bonobo females experience extended periods of estrus compared to female chimpanzees. This, in turn, attracts many male bonobos to them, creating large subgroups or parties in the process. Female bonobos, through an unspecified mechanism (see lines 135 – 142 this manuscript), are also attracted to these mixed-sex parties containing other estrous females, leading to increased gregariousness among female bonobos.

The authors ostensibly test these two hypotheses, which they admit are not mutually exclusive, by collecting comparative data on the grouping behavior of bonobos and chimpanzees. They make several

predictions based on what they call the “female ecology hypothesis” (lines 193 – 201) and “female sexuality hypothesis” (lines 208 – 217). Their findings indicate that:

- 1) Female bonobos are more gregarious, as measured by female party size, than female chimpanzees. This difference, however, vanishes when controlling for the number of females in bonobo and chimpanzee groups and the presence of estrus bonobo and chimpanzee females.
- 2) The percentage time spent alone by female bonobos and chimpanzees did not differ nor did the amount of time females in each species spent with each other.

One other hidden result seems to indicate that:

- 3) Bonobo females are in estrus more frequently than are chimpanzee females. This is suggested by the findings that maximally tumescent bonobo females were present much more often than maximally tumescent chimpanzee females and that the average number of the former in parties was three times higher than the number of the latter. A direct tally of how often females in each species spent in estrus is not provided, however.

Information on how often females are maximally tumescent in each of our study group has now been added to the revised version of the manuscript.

From these three findings, the authors conclude that extended sexual signaling accounts for species differences in grouping patterns between bonobos and chimpanzees. They go further, though, and make a bold claim that these results also explain why female bonobos are socially dominant to male conspecifics and why the latter do not sexually coerce the former.

This is an interesting paper, but there are several problems. These problems involve the paper’s formulation, analysis, and interpretations. First, the paper is set up as a test of two competing hypotheses, essentially whether food or sex, influence bonobo and chimpanzee grouping patterns. At the outset we need data on the food supply for bonobos and chimpanzees to adequately test the female ecology hypothesis. None are provided, however.

Following the reviewer’s comments, the revised version of our manuscript has now a more focused hypotheses framework, testing exclusively for proposed differences in female-female grouping patterns and underlying processes. The revised models are more tailored to the predictions and control for fluctuations in food availability in several ways, including a carefully validated use of a behavioural proxy (see details in the reply to the comments from reviewer 1). This addition allows us to test for the robustness of the main findings on the influence of female sexual swellings on grouping patterns, even after controlling for ecological confound. As mentioned above our results hold even after including food abundance in our statistical models.

Alternatively, to test the female sexuality hypothesis, we require information on how often female bonobos and chimpanzees spend in estrus. As noted above, we are not furnished direct information on this matter. An additional problem is that, as the authors themselves admit (lines 145 – 146), the two hypotheses are not mutually exclusive, and the factors that influence the feeding ecology hypothesis might ultimately affect the reasons why extended sexual signaling in bonobos might exist. A more illuminating paper would have made an attempt to disentangle these, and with that, provide a stronger test of the two hypotheses.

We add the information of how often focal females are in estrus and how often estrous females are in the focal party in the revised version of the manuscript (Line 424ff). We also controlled for ecological variation to test for species differences in underlying grouping processes linked to sexual swellings in order to disentangle the processes more, as suggested by the reviewer.(Line 431ff)

Analytically, data on the grouping behavior of bonobos and chimpanzees is used to test the two hypotheses. But as indicated above, grouping data alone is insufficient to test them. One of the main results that female bonobos are more gregarious than female chimpanzees appears to be consistent with the female ecology hypothesis, but this is tempered by the finding that this difference vanishes when one controls for the presence of estrus females. The significance of this result is questionable, as the authors themselves admit, their analysis uses a subset of the dependent variable as an independent variable.

In the revised version of the manuscript we account for the presence of maximally tumescent females in two new different ways as suggested by reviewer 1. This has the advantage now to remove the dependence between the response variable and the test variable. Specifically, we included the presence of maximally tumescent female as a categorical variable (0, one, and more than 1) (as opposed to a continuous actual number of maximally tumescent female before), as well as, in a different analysis, the percentage of time during a focal follow during which a maximally tumescent female was present in the party. Both approaches reveal comparable results (see above) and highlight that species differences in female-female gregariousness disappear in the absence of maximally tumescent females.

Another important finding that the difference in female grouping patterns disappears when one controls for the number of females in each species is interesting, but without information about the food supply, one can't evaluate it. Similar feeding conditions for the two species over the relatively short duration of study, a possibility consistent with the feeding ecology hypothesis, might explain the lack of difference.

Following the comments from reviewer 1 we now focus on the number of females in the party and not on the % of the community females present. However, to rule out that the results are driven by ecological fluctuations, in the revised version of the manuscript we include a carefully validated behavioural proxy for food availability (time spent feeding, see reply to reviewer 1 comments for more details). We further validated these results by running the analysis on a smaller dataset for which we have actual data on variation in fruit availability which led to similar results as the model with all the observation data. By including these ecological factors as control predictors in our models, we can make statement about the influence of estrous females on grouping pattern independent of food availability.

Finally, the authors go far beyond the data they present and argue that their findings indicate why female bonobos are socially dominant to male conspecifics and why the latter do not sexually coerce the former as male chimpanzees do to their female conspecifics. No data are provided on either of these points, however. The authors note (lines 549-552) that we require more information on male aggression toward females, female-male dominance relationships, and female sexuality to make such conclusions. None are provided. And in the next sentence, they admit that the evolutionary origin of female cooperation in bonobos that leads to their ability to dominate males remains "puzzling."

We make it clear in the revised version of the manuscript that we focus on differences in female-female gregariousness and move all the speculative parts about the implications for cooperation and male competition into the discussion. This helps clarifying the framing and scope of our study.

In sum, this paper contains some interesting comparative data regarding the grouping behavior of bonobos and chimpanzees. What to make of these data, though, is difficult to discern.

We acknowledge the reviewer's doubts. We believe that the focus on female data, the inclusion of a proxy for food availability in our analysis and a streamlined introduction in the revised version of the manuscript helps a lot with clarifying our results and interpretation.

Citations

- de Waal, F. B. M. (1997). *Bonobo, The forgotten ape*. University of California Press.
- Hanya, G. (2004). Diet of a Japanese macaque troop in the coniferous forest of Yakushima. *Int J Primatol*, 25, 55–71.
- Harris, T. R., Chapman, C. A., & Monfort, S. L. (2010). Small folivorous primate groups exhibit behavioral and physiological effects of food scarcity. *Behavioral Ecology*, 21(1), 46–56.
<https://doi.org/10.1093/beheco/arp150>
- Knott, C. D. (2005). Energetic responses to food availability in the great apes: Implications for hominin evolution. In D. K. Brockman & C. P. Van Schaik (Eds.), *Seasonality in primates: Studies of living and extinct human and non-human primates* (pp. 351–378). Cambridge University Press.
- Masi, S., Cipolletta, C., & Robbins, M. M. (2009). Western Lowland Gorillas (*Gorilla gorilla gorilla*) Change Their Activity Patterns in Response to Frugivory. *American Journal of Primatology*, 71(2), 91–100. <https://doi.org/10.1002/ajp.20629>
- Mulavwa, M., Furuichi, T., Yangozene, K., Yamba-Yamba, M., Motema-Salo, B., Idani, G., Ihobe, H., Hashimoto, C., Tashiro, Y., & Mwanza, N. (2008). Seasonal Changes in Fruit Production and Party Size of Bonobos at Wamba. In T. Furuichi & J. Thompson (Eds.), *The Bonobos* (pp. 121–134). Springer New York.
- Valé, P. D., Béné, J.-C. K., N'Guessan, A. K., Crockford, C., Deschner, T., Koné, I., Girard-Buttoz, C., & Wittig, R. M. (2020). Energetic management in wild chimpanzees (*Pan troglodytes verus*) in Taï National Park, Côte d'Ivoire. *Behavioral Ecology and Sociobiology*, 75(1), 1.
<https://doi.org/10.1007/s00265-020-02935-9>
- Watts, D. P. (1988). Environmental influences on mountain gorilla time budgets. *American Journal of Primatology*, 15(3), 195–211. <https://doi.org/10.1002/ajp.1350150303>

Reviewers' comments:

Reviewer #4 (Remarks to the Author):

My role as a reviewer is limited by the fact that I have only seen this second, much revised version of the manuscript and that I have been asked to focus on how the reviewers' input has been addressed in this revision.

As far as I can tell, the reviewers' comments to the original submission did help the authors to strongly tighten and focus their analyses. Large aspects were removed, the hypotheses were clarified and narrowed down, and additional data used to support certain aspect. Overall, this looks like a thorough revision and a great improvement.

I have the following few minor comments of my own to add:

-- It is unfortunate, that there are no direct measures of food availability, that feeding time had to be used as proxy. I hope that, moving forward, this will change (also for studies on apes) especially for topics such as the one addressed in this manuscript.

-- Do you not expect seasonal effects? The study periods do not cover all months of the year.

-- Would it be a good idea to remove the only nulliparous female from the analysis? She was also observed for the shortest duration (4 hours, the next shortest is already 4 times longer).

-- It sounds in the methods as if one observer was at two places at the same time (Feb – May 2016).

-- The grouping for the number of MTF is unclear. In the Figures it is given as zero, <1, and >1. What is the difference between zero and <1? In which group do cases with exactly 1 female fall? You might want to clarify this in the text, the Figures and the Tables (e.g., 0, <1, >1 or is it 0, 1, >1 ?)

-- The caption of Table 2 is incomplete

-- Typo in the captions of SI Table 1. The table headers contain unexplained abbreviations

-- line 700: muriquis, not 'muriquis'

-- In SI Figure 2 the species in the legend are swapped

-- References

1) Some titles have capital letters, most don't.

2) Some species names are not in italic

3) The position of the Publisher for book chapters varies

4) Sometimes the editor(s) are not provided

5) There are typos, e.g., "Cambrdge"

6) Occasionally a doi is provided, mostly not.

Reviewer #5 (Remarks to the Author):

This paper examines the proximate drivers of differences in gregariousness in female bonobos and chimpanzees, with the aim of explaining the resulting differences between both species in female social behavior. The study is based on a well-suited dataset, as data were collected on both species following the same protocol, and focal individual observations have been used to represent as accurately as possible time spent alone vs in groups. Although I didn't read the original manuscript, I think the decisions in the revised manuscript to focus only on female grouping patterns and to control for fluctuations in food availability greatly improve the paper, and address some of the issues raised by the reviewers. Still, I think the paper's conceptual framework would need some restructuring, so that there is a better match between the actual results and the introduction and discussion. I raise the main issues below, in the hope they help the authors improve the clarity and the message of this paper, as I really believe the results are interesting and raise important questions for further research.

The key problem to me is that the introduction of the 'ecology process' and the 'sexual signaling attraction process' is confusing, which has repercussions throughout the whole manuscript.

- A first paragraph (L83 ff) introduces the classic socio-ecological theory, that states that higher food availability lowers feeding competition, and therefore allows for the development of stronger relationships. Then there is a shift to explaining two 'processes driving the variation in sociality in

chimpanzees and bonobos'. It isn't clear to me how these 'processes' relate to the existing hypotheses (the socio-ecological model and female sexual signaling models), and why the socio-ecology model is used as some kind of introduction to these 'processes'. It seems that the 'ecology process' is a repetition of the socio-ecological theory, whereas the 'sexual signaling attraction process' is rather an extension of female sexual signal models, where the difference in length and frequency of swellings (which is the part explained by the female sexual signal models) is linked to differences in gregariousness (which is not a part of those original models). This is confusing as throughout the paper, the 'ecology process' and the 'sexual signaling attraction process' are discussed as if they are accepted theories that can explain differences in gregariousness between chimps and bonobos. These processes, however, are ideas based on existing theories, with derived predictions that do not test those theories explicitly. A clearer set-up of what exactly the hypotheses tested in this paper are, and how they relate to existing theories would allow for a better framing of the results and interpretation of their meaning for differences in social behavior.

- I also find there is some circularity in the reasoning, particularly with regards to the 'sexual signaling attraction process'. The paper sets out by introducing behavioral differences between bonobos and chimpanzees: female dominance, less male aggression towards females and more female-female alliances in bonobos than in chimps (L63-82). It then puts the difference in female swelling duration & frequency forward as a potential driver for these behavioral differences, especially since fully swollen females might attract females more/directly in bonobos because of differences in their social behavior, such as lower risk of male aggression and female dominance (L112-126). So, what came first: the differences in social structure, the differences in the attractiveness of fully swollen females, or is this a self-reinforcing process? More generally: is it increased female gregariousness that allows for stronger, more cooperative relationships in bonobos, or is it the nature of the social relationships in bonobos that permit the formation of larger female parties? E.g., L89: 'There are two described potential processes driving the variation in sociality in chimpanzees and bonobos, potentially resulting in species difference in female associations.' seems to imply that sociality drives associations, whereas L 178 'If current evolutionary mechanisms that promote higher female-female gregariousness in bonobos than in chimpanzees drive the variation observed in bonobo and chimpanzee social behavior' suggests it is the other way around.

- Linked to this, I am not sure how exactly the authors link the 'sexual signaling attraction process' to differences in female social behavior and cooperation between chimps and bonobos. In L102, the 'sexual signaling attraction process' is introduced as 'A second process is based on the effects of females sexual signaling on variation in female gregariousness.', so that the idea is that differences in sexual signaling drive differences in gregariousness, which, in turn, drive differences in cooperation. But then in the discussion (L555 ff) 'Given the findings of very small numerical differences in overall female-female gregariousness, it seems unlikely that current selection favoring a generally higher female affinity in bonobos is ultimately driving the proposed species differences in female cooperation [...]. The significant species differences in female-female gregariousness match differences in sexual signaling and in the female affinity to potentially fertile females. Therefore, our results support the idea that changes in female signaling are a proximate driver of species differences in social structures [...].', the idea seems to be that it is the affinity for fertile females per se that predicts differences in cooperation, so not through a step of increased gregariousness. Which also seems to be the message in the conclusion (L709 ff) '[...] that some marked species differences in female-female gregariousness arise from variation in female sexual signaling that in itself could explain some of the observed behavioral differences.' This issue is exacerbated by the authors using gregariousness, cohesiveness and affinity seemingly interchangeably, so that it isn't clear whether they refer to the formation of larger groups or to the relationships formed within those groups.

Minor comments:

- L178 ff: Why would we expect more females in female parties and females to spend less time alone in bonobos if current evolutionary mechanisms that promote female gregariousness drive the variation observed in bonobo and chimpanzee behavior? I suppose what is meant is that because bonobo females form more cooperative behaviors, one would expect them to be more

gregarious, if indeed gregariousness predicts social behaviors. This could be stated more explicitly. More importantly, I don't think that simply finding that female bonobos are more gregarious proves that indeed gregariousness drives the difference in social behaviors. I also think it's confusing that '1) a larger number of females in female focal parties' and '2) less time spent alone by females' aren't predictions like the ones below (P1a, P1b, P2.), especially since both do get tested (Model 1, L288 ff).

- L304: It would be nice to report the range of MTF in each community, so the reader has an idea of what variation gets collapsed into the >1 category.

Reviewers' comments:

Reviewer #4 (Remarks to the Author):

My role as a reviewer is limited by the fact that I have only seen this second, much revised version of the manuscript and that I have been asked to focus on how the reviewers' input has been addressed in this revision.

As far as I can tell, the reviewers' comments to the original submission did help the authors to strongly tighten and focus their analyses. Large aspects were removed, the hypotheses were clarified and narrowed down, and additional data used to support certain aspect. Overall, this looks like a thorough revision and a great improvement.

We thank the reviewer for this overall very positive evaluation of our revisions

I have the following few minor comments of my own to add:

-- It is unfortunate, that there are no direct measures of food availability, that feeding time had to be used as proxy. I hope that, moving forward, this will change (also for studies on apes) especially for topics such as the one addressed in this manuscript.

We agree with the reviewer and add a statement in the discussion echoing the reviewers point :

(L650) "For future studies the inclusion of direct measures of food availability would be preferable"

-- Do you not expect seasonal effects? The study periods do not cover all months of the year.

Generally, we do expect seasonal effects in the occurrences of sexual swellings, but they are most likely liked to change in food availability (both species are not seasonal breeders). By controlling for the ecology, we therefore account for potential seasonal effects. Beyond the influence of the ecology, we do not see a biological mechanism by which the influence of sexual swellings itself on the number of female party members varies seasonally.

To address the reviewer's point, we add in the revised version a graph (SI Figure 4) showing the changes in the number of female party members through our study period.

-- Would it be a good idea to remove the only nulliparous female from the analysis? She was also observed for the shortest duration (4 hours, the next shortest is already 4 times longer).

We agree with the reviewer's suggestion. We thus rerun all the analyses without the nulliparous females in the revisions which did not change the result. In the revised version we present only the data from parous females (accordingly we redid all tables and figures representing the data)

-- It sounds in the methods as if one observer was at two places at the same time (Feb – May 2016).

We clarified this by adding that CGB divided time between the two sites (i.e he collected data at one site while assistants collected data at the other site at the same time)

(L254) "(with CGB dividing time between the two sites) "

-- The grouping for the number of MTF is unclear. In the Figures it is given as zero, <1, and >1. What is the difference between zero and <1? In which group do cases with exactly 1 female fall? You might want to clarify this in the text, the Figures and the Tables (e.g., 0, <1, >1 or is it 0, 1, >1 ?)

We clarified this in the respective figures and figure captions by including the information from the methods which states that: "0 (no maximally tumescent female [including focal swelling] individual present), <=1 (average of 1 or less but not 0 maximally tumescent females [including focal swelling] present during a given focal follow), >1 (average of more than 1 maximally tumescent female [including focal swelling] present during the focal follow)."

L324/Figure 2

-- The caption of Table 2 is incomplete

We added the missing period

-- Typo in the captions of SI Table 1. The table headers contain unexplained abbreviations

We corrected the typo and added the meaning of MTF

-- line 700: muriquis, not 'muriquis'

This has been corrected

-- In SI Figure 2 the species in the legend are swapped

This has been corrected

-- References

1) Some titles have capital letters, most don't.

This has been corrected

2) Some species names are not in italic

ALL species names are now italic

3) The position of the Publisher for book chapters varies

This has been unified

4) Sometimes the editor(s) are not provided

This has been corrected

5) There are typos, e.g., "Cambrdge"

This has been corrected

6) Occasionally a doi is provided, mostly not.

After clarifying with the journals requirements, we removed the doi (as all the citations that are not only online material)

Reviewer #5 (Remarks to the Author):

This paper examines the proximate drivers of differences in gregariousness in female bonobos and chimpanzees, with the aim of explaining the resulting differences between both species in female social behavior. The study is based on a well-suited dataset, as data were collected on both species following the same protocol, and focal individual observations have been used to represent as accurately as possible time spent alone vs in groups. Although I didn't read the original manuscript, I think the

decisions in the revised manuscript to focus only on female grouping patterns and to control for fluctuations in food availability greatly improve the paper, and address some of the issues raised by the reviewers. Still, I think the paper's conceptual framework would need some restructuring, so that there is a better match between the actual results and the introduction and discussion. I raise the main issues below, in the hope they help the authors improve the clarity and the message of this paper, as I really believe the results are interesting and raise important questions for further research.

We thank the reviewer for an overall positive evaluation of our revisions

The key problem to me is that the introduction of the 'ecology process' and the 'sexual signaling attraction process' is confusing, which has repercussions throughout the whole manuscript.

- A first paragraph (L83 ff) introduces the classic socio-ecological theory, that states that higher food availability lowers feeding competition, and therefore allows for the development of stronger relationships. Then there is a shift to explaining two 'processes driving the variation in sociality in chimpanzees and bonobos'. It isn't clear to me how these 'processes' relate to the existing hypotheses (the socio-ecological model and female sexual signaling models), and why the socio-ecology model is used as some kind of introduction to these 'processes'. It seems that the 'ecology process' is a repetition of the socio-ecological theory, whereas the 'sexual signaling attraction process' is rather an extension of female sexual signal models, where the difference in length and frequency of swellings (which is the part explained by the female sexual signal models) is linked to differences in gregariousness (which is not a part of those original models). This is confusing as throughout the paper, the 'ecology process' and the 'sexual signaling attraction process' are discussed as if they are accepted theories that can explain differences in gregariousness between chimps and bonobos. **These processes, however, are ideas based on existing theories, with derived predictions that do not test those theories explicitly.** A clearer set-up of what exactly the hypotheses tested in this paper are, and how they relate to existing theories would allow for a better framing of the results and interpretation of their meaning for differences in social behavior.

We agree with the reviewer that the "ecological" and the "sexual signaling attraction process" are derived ideas from other hypotheses and in our manuscript, we do not test these hypotheses directly. In the previous round of reviews, we have been reminded by 2 reviewers that the 'sexual signaling attraction process' does not exclude the idea that ultimately it is differences in ecology that "allow" females to change their sexual signaling and that socioecological theory stand behind both. In the revisions we therefore tried to clarify the level of processes that we are testing and emphasized that we test for differences in the proximate processes of changes in female gregariousness between the species.

In the second round of revisions, we now restructured the introduction to link the "ecology process" more directly to socioecological theory and acknowledge the closer link between ecological processes and socioecological theory.

(L86ff) "One hypothesized process that drives the variation in sociality in chimpanzees and bonobos and potentially results in species difference in female associations, is directly related to this framework and emphasizes only aspects of the ecology. According to this first hypothesized process, a more.."

We also emphasize a closer link of the "sexual signaling attraction process" to models of sexual signaling.

(L100ff) “A second hypothesized process, that potentially drives the variation in sociality in chimpanzees and bonobos, is closer linked to theories on sexual signaling that explain aspects of sexual swellings in primates^{52,53}. This second hypothesized process relates differences in sexual swellings between the species to variation in behaviour and consequently to female-female gregariousness.”

Furthermore, we make clear that we do not test the broad theories per se, but rather the derived proximate processes for species differences in gregariousness between chimpanzees and bonobos.

(L138) “Understanding how these two hypothesized processes account for variation in female sociality within and between species potentially allows us to identify proximate processes that maintain the observed species differences in female..”

(L192) “To specifically test whether the generally more favorable ecological conditions in bonobos are the main drivers of the hypothesized species difference in female-female gregariousness between chimpanzees and bonobos (ecology process),...”

- I also find there is some circularity in the reasoning, particularly with regards to the ‘sexual signaling attraction process’. The paper sets out by introducing behavioral differences between bonobos and chimpanzees: female dominance, less male aggression towards females and more female-female alliances in bonobos than in chimps (L63-82). It then puts the difference in female swelling duration & frequency forward as a potential driver for these behavioral differences, especially since fully swollen females might attract females more/directly in bonobos because of differences in their social behavior, such as lower risk of male aggression and female dominance (L112-126). So, what came first: the differences in social structure, the differences in the attractiveness of fully swollen females, or is this a self-reinforcing process? More generally: is it increased female gregariousness that allows for stronger, more cooperative relationships in bonobos, or is it the nature of the social relationships in bonobos that permit the formation of larger female parties? E.g., L89: ‘There are two described potential processes driving the variation in sociality in chimpanzees and bonobos, potentially resulting in species difference in female associations.’ seems to imply that sociality drives associations, whereas L 178 ‘If current evolutionary mechanisms that promote higher female-female gregariousness in bonobos than in chimpanzees drive the variation observed in bonobo and chimpanzee social behavior’ suggests it is the other way around.

We see the reviewer point and tried to make the argument less circular in the revised version. After careful consideration we think that the ecological process argues along “regariousness first” while the “sexual signaling attraction process” relates more to a self-reinforcing process where the nature of the relationship between females has to be a given way to result in benefits of increased female-female gregariousness. Given our finding of very weak overall differences in female gregariousness, we are inclined to think that changes in the nature of the relationships rather than changes in overall gregariousness are relevant in understanding underlying causes of species differences in behavioural domains. In the introduction of the revised version of the manuscript we try to be more explicit about this reasoning.

Ecology process:

(L91)“This would result in a generally higher female-female gregariousness in bonobos by the virtue of ecology alone (assuming comparable benefits of sociality) allowing for the establishment of stronger bonding and cooperation among females. “

Sexual signaling attraction process:

(L102)“This second hypothesized process relates differences in sexual swellings cycles between the species to variation in behaviour and consequently to female-female gregariousness.”

(L122) “These benefits are specific to the nature of the social relationships observed in bonobos (impactful mother-son bonds and female-female cooperation facilitated by sociosexual behaviour), and likely result in self-reinforcing processes driving potentially higher females’ gregariousness in this species.”

(L131) “Again, these benefits are closer linked to the nature of the social relationships in bonobos and not necessarily to a generally higher female-female gregariousness.”

Generally, our findings of very little actual differences in overall female-female gregariousness points at a scenario in which changes in the relationships (likely tied to changes in sexual signaling) are the crucial starting point of a self-reinforcing process and that the evidence we present rejects the idea that “current evolutionary mechanisms that promote higher female-female gregariousness in bonobos than in chimpanzees drive the variation observed in bonobo and chimpanzee social behavior”.

In the revised version of the manuscript we 1) state that ecology process assumes a higher gregariousness as the starting point for species differences in the nature of the relationships, and 2) emphasize that the additional benefits of associating with MTF are specific to the nature of the social relationships observed in bonobos and likely result in self-reinforcing processes driving potentially higher females’ gregariousness in this species.

See above

- Linked to this, I am not sure how exactly the authors link the ‘sexual signaling attraction process’ to differences in female social behavior and cooperation between chimps and bonobos. In L102, the ‘sexual signaling attraction process’ is introduced as ‘A second process is based on the effects of females sexual signaling on variation in female gregariousness.’, so that the idea is that differences in sexual signaling drive differences in gregariousness, which, in turn, drive differences in cooperation. But then in the discussion (L555 ff) ‘Given the findings of very small numerical differences in overall female-female gregariousness, it seems unlikely that current selection favoring a generally higher female affinity in bonobos is ultimately driving the proposed species differences in female cooperation [...]. The significant species differences in female-female gregariousness match differences in sexual signaling and in the female affinity to potentially fertile females. Therefore, our results support the idea that changes in female signaling are a proximate driver of species differences in social structures [...].’, the idea seems to be that it is the affinity for fertile females per se that predicts differences in cooperation, so not through a step of increased gregariousness.

Which also seems to be the message in the conclusion (L709 ff) '[...] that some marked species differences in female-female gregariousness arise from variation in female sexual signaling that in itself could explain some of the observed behavioral differences.' This issue is exacerbated by the authors using gregariousness, cohesiveness and affinity seemingly interchangeably, so that it isn't clear whether they refer to the formation of larger groups or to the relationships formed within those groups.

We tried to be more specific in the revised version and emphasize that 1) the affinity for fertile females under the 'sexual signaling attraction process' results from benefits of MTF that are in the nature of the cooperative relationships (see above) and 2) that differences in cooperation do not go through a step of increased gregariousness, or more accurately: do not depend at this point in time on a higher gregariousness of bonobo females (we do not address the specific process leading to female-female coalitions, but can refute the idea that it originates from an generally increased gregariousness).

(L676)“Our results seem to indicate that an increased general cohesion and attraction between females might not be the underlying species difference facilitating stronger female alliance formation in the first place, as these differences are neither very pronounced nor persisting in the absence of sexual signaling..”

Finally, we unified the terms gregariousness/cohesiveness and affinity by removing the term “cohesiveness” and specifying that the term affinity refers to affinity in parties.

Minor comments:

- L178 ff: Why would we expect more females in female parties and females to spend less time alone in bonobos if current evolutionary mechanisms that promote female gregariousness drive the variation observed in bonobo and chimpanzee behavior? I suppose what is meant is that because bonobo females form more cooperative behaviors, one would expect them to be more gregarious, if indeed gregariousness predicts social behaviors. This could be stated more explicitly. More importantly, I don't think that simply finding that female bonobos are more gregarious proves that indeed gregariousness drives the difference in social behaviors. I also think it's confusing that '1) a larger number of females in female focal parties' and '2) less time spent alone by females' aren't predictions like the ones below (P1a, P1b, P2..), especially since both do get tested (Model 1, L288 ff).

In the revised version we try to be more specific in the wording and adopt the reviewer's suggestion of how to state is explicitly. Furthermore, we attribute prediction numbers, as suggested to the mentioned predictions.

(L187)“ If a generally higher gregariousness in bonobos than in chimpanzees promotes the differences in social behaviour, including patterns of female cooperation, aggression and dominance between the sexes (as hypothesized by the ecology process), we would expect 1) a larger number of females in parties of focal females (Prediction P1) and 2) less time spent alone by females in LuiKotale bonobos compared to Tai chimpanzees (Prediction P2).”

- L304: It would be nice to report the range of MTF in each community, so the reader has an idea of what variation gets collapsed into the >1 category.

We added this information in the revised version of the manuscript by stating in the method section the maximal number of MTF that were seen present in a party.

(L327)“The maximal number of maximally tumescent females present in the party was 8 in bonobos and 3 in both chimpanzee communitiess.”